# Generated Contents Enrichment

## Abstract

In this paper, we investigate a novel artificial intelligence generation task termed Generated Contents Enrichment (GCE). Conventional AI content generation produces visually realistic content by implicitly enriching the given textual description based on limited semantic descriptions. Unlike this traditional task, our proposed GCE strives to perform content enrichment explicitly in both the visual and textual domains. The goal is to generate content that is visually realistic, structurally coherent, and semantically abundant. To tackle GCE, we propose a deep end-to-end adversarial method that explicitly explores semantics and inter-semantic relationships during the enrichment process. Our approach first models the input description as a scene graph, where nodes represent objects and edges capture inter-object relationships. We then adopt Graph Convolutional Networks on top of the input scene description to predict additional enriching objects and their relationships with the existing ones. Finally, the enriched description is passed to an image synthesis model to generate the corresponding visual content. Experiments conducted on the Visual Genome dataset demonstrate the effectiveness of our method, producing promising and visually plausible results.

## 1 Introduction

Given the rich semantics and complex inter-semantic relationships in the real-world scenes in Fig. 1(f), their available descriptions, shown in Fig. 1(a), are usually inadequately informative, thus leading to the semantic richness gap between the accordingly generated images and the real-world ones, as shown in Fig. 1(d) and (f). However, given the simple descriptions, our human minds can easily hallucinate a semantically abundant image, filling the semantic richness gap by first understanding the given description, then enriching it with its related semantics and inter-semantic relationships, and finally hallucinating the accordingly generated image.

We term this task, barely investigated in conventional research, as generated contents enrichment (GCE). Although the human brain can fulfill this task effortlessly, existing artificial intelligence content generation methods (Karras et al., 2020; Vahdat & Kautz, 2020; Chen et al., 2020; Saharia et al., 2022; Rombach et al., 2022; Esser et al., 2021; Yang et al., 2022c; Herzig et al., 2020) produce results with inadequate semantic richness. In contrast to our approach, these methods overlook explicit content enrichment by neglecting reasoning on relevant semantics and inter-semantic relationships. As shown in Fig. 1(d), the state-of-the-art (SOTA) generation approach (Rombach et al., 2022), without reasoning the related semantics of the given descriptions, generates the images with insufficient semantics compared with the real-world ones in Fig. 1(f). Indeed, existing content generation methods, to the best of our knowledge, primarily rely on implicit learning to generate visual contents beyond those explicitly described in the input. However, this approach overlooks the semantic richness and structural reasonableness, making these methods inadequate for addressing GCE.

Although GCE is essential for filling the semantic richness gap between the short description and the expected image, it is challenging to learn to solve it. Firstly, the enriching content should be semantically relevant to the ones in the description and form appropriate inter-semantic relationships. Secondly, the enriched scene graph, composited by the description and its enriched semantics, should be structurally real, following the distribution of scene graphs from real-world images.

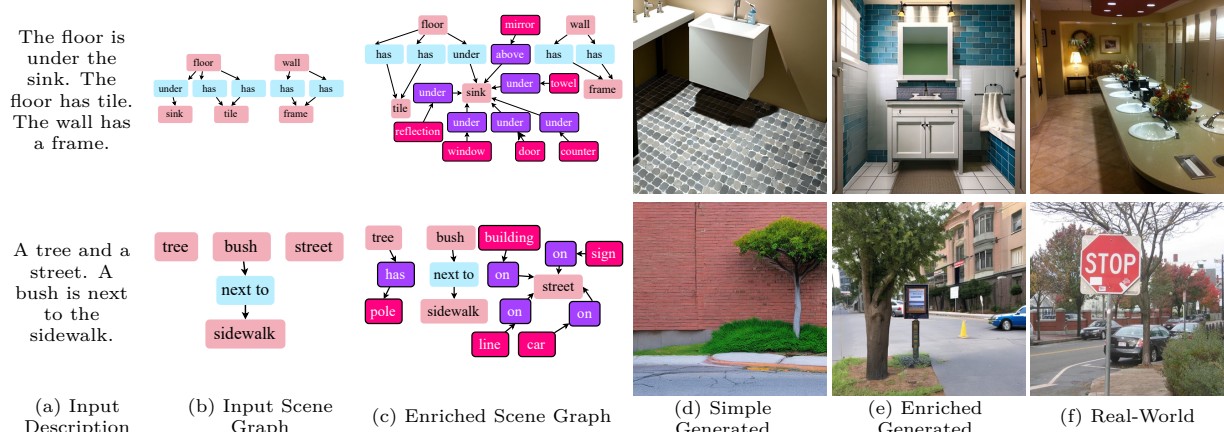

(a) Input Description · (b) Input Scene Graph · (c) Enriched Scene Graph · (d) Simple Generated · (e) Enriched Generated · (f) Real-World

Figure 1: Initially, the input textual description (a) is represented as a scene graph (b). Subsequently, we enrich the scene description (b) iteratively by appending additional objects and their relationships with existing scene elements. The enriching content (c) should preserve the same essential scene characteristics as the original input (b). This enrichment is executed utilizing our proposed end-to-end adversarial graph convolutional framework. Both the input (b) and enriched (c) scene graphs are then employed to synthesize simple (d) and enriched (e) images, respectively. In comparison to the simple image (d), the enriched one (e) not only reflects the essence of the initial input description (b) but also integrates more relevant intricate details akin to those present in real-world images (f) found in the Visual Genome dataset (Krishna et al., 2017).

Towards solving the mentioned challenges and achieving GCE, we propose an intriguing approach that, to some extent, mimics the human reasoning procedure for hallucinating the enriching content. Particularly, our introduced method is designed with an end-to-end trainable architecture consisting of three principal stages. As an initial step, we represent the input description as a scene graph, wherein each node resembles a semantic, i.e., an object, and each edge corresponds to an inter-object relationship. In the first stage, we enrich the scene graph iteratively. In each iteration, we append a single object, representing it as an unknown node, along with its inter-object relationships as unknown edges. These are then predicted by employing a Graph Convolutional Network (GCN), which aggregates information from known nodes and edges. As a result, the enriching contents are semantically aligned with the ones in the input description. Then, the enriched scene graph is fed into a pair of Scene Graph Discriminators to guarantee its structural reality. In the second stage, we synthesize the enriched image from the enriched scene description by adopting an image generator (Johnson et al., 2018; Rombach et al., 2022). The third stage simultaneously feeds the enriched image and the input description into the Visual Scene Characteriser and Image-Text Aligner, thus the generated image is visually real and preserves the information of the description.

Therefore, our contribution is the introduction of a novel adversarial end-to-end trainable framework designed to address the proposed GCE task as the first attempt, to our best knowledge. The GCE task aims to bridge the semantic richness gap between human hallucination and SOTA image synthesizers. Traditional text-to-image generation approaches solely focus on implicit visual enrichments. However, we enhance the generation by incorporating both explicit textual and visual content enrichments, ensuring semantic and structural coherence with the given input description. Our approach guarantees that the final enriched image accurately reflects the key scene elements specified in the input. We evaluate our method using real-world images and richly annotated scene descriptions from the Visual Genome dataset (Krishna et al., 2017), demonstrating promising quantitative and qualitative results.

## 2 Related Work

### 2.1 Realistic Image Generation

Generative Adversarial Networks (GANs) (Goodfellow et al., 2020; Radford et al., 2015) jointly train a generator and discriminator to distinguish real from fake synthesized images. GANs produce images of high

quality that are, in some cases, almost impossible for humans to discriminate from real images (Brock et al., 2018; Karras et al., 2020; Mirza & Osindero, 2014; Odena et al., 2017). However, their training procedure is not as stable as the other methods, and they may not model all parts of the data distribution (Metz et al., 2016). The first attempts for GAN-based text-to-image synthesis (Reed et al., 2016a;b) are followed by stacking GANs (Zhang et al., 2018) and employing the attention mechanism (Xu et al., 2018; Bahdanau et al., 2014; Vaswani et al., 2017). Variational Auto Encoder (VAE) (Kingma & Welling, 2013) consists of an encoder to a latent space and a decoder to return to the image space. VAE is also employed for image synthesis (Vahdat & Kautz, 2020; Huang et al., 2018), generating high-resolution data but with lower quality than their GAN counterpart. Alternatively, autoregressive approaches (Van Den Oord et al., 2016; Van den Oord et al., 2016; Chen et al., 2020; Child et al., 2019; Yu et al., 2022) synthesize pixels conditioned on an encoded version of the input description that could depend on the previously generated pixels with computationally expensive training and sequential sampling process. Although SOTA diffusion-based (Sohl-Dickstein et al., 2015) models (Dhariwal & Nichol, 2021; Ho et al., 2020; Song et al., 2020; Saharia et al., 2022) synthesize images of astounding quality, these architectures necessitate elevated computational costs during training and a time-consuming inference. Stable Diffusion (Rombach et al., 2022) is one of the first open-source methods that attracted so much attention among artists as it applies the diffusion process to low-dimensional image embeddings. On the other hand, there has been some research on a hybrid two-stage method like VQ-VAEs (Razavi et al., 2019; Yan et al., 2021) and VQGANs (Esser et al., 2021; Yu et al., 2021) which aims to benefit from a mixed version of the mentioned methods.

## 2.2   Scene Graphs

Scene graphs proposed in Johnson et al. (2015a) and further investigated in Tripathi et al. (2021); Rosinol et al. (2021); Zhang et al. (2019) are a structured version of a scene representation containing information about objects, their relationships, and often their attributes. Previously, understanding a scene was limited to detecting and recognizing the objects in an image. However, scene graphs aim to obtain a higher level of scene understanding (Chang et al., 2021) and are employed to improve action recognition (Aksoy et al., 2010), image captioning (Aditya et al., 2015), and image retrieval (Johnson et al., 2015b). Scene graphs may be predicted from images (Lu et al., 2016; Xu et al., 2017; Yang et al., 2018; Tang et al., 2020; Yang et al., 2022b; Li et al., 2018) or serve as inputs to generate synthesized images (Johnson et al., 2018; Yang et al., 2022c; Herzig et al., 2020; Li et al., 2019; Ashual & Wolf, 2019).

## 2.3   Molecular Generation

Molecular generative models (Simonovsky & Komodakis, 2018; You et al., 2018; Shi et al., 2020; Luo et al., 2021; De Cao & Kipf, 2018; Goyal et al., 2020; Grover et al., 2019; Jin et al., 2018; Samanta et al., 2020) are designed specifically for molecular graphs, with a focus on applications in drug discovery. These models aim to generate molecular graphs that adhere to chemically valid structures while satisfying specific chemical property constraints. However, they are not directly applicable to scene graphs due to significant differences in structure. Molecular graphs are extremely smaller and less diverse, featuring fewer node types and relationships compared to scene graphs. Additionally, molecular generative models often assume that the input graph is connected, a property that does not necessarily hold for scene graphs (Agarwal et al., 2023).

## 2.4   Semantic Expansion

Our work introduces and tackles the task of GCE, which aims to enhance the quality of image synthesis. While prior research has explored various forms of semantic expansion, their goals, input-output structures, and evaluation criteria differ significantly from ours.

The task of visual comprehension is addressed by generating an enriched and expressive scene graph as output from a given input image (Khan et al., 2023; 2022). These approaches incorporate common-sense knowledge derived from the input image to enhance the scene graph. Additionally, Yang et al. (2022a) focuses on the task of image outpainting, where an input image is extrapolated beyond its original boundaries by constructing and extending scene graphs based on the image. While these works are centered on generating

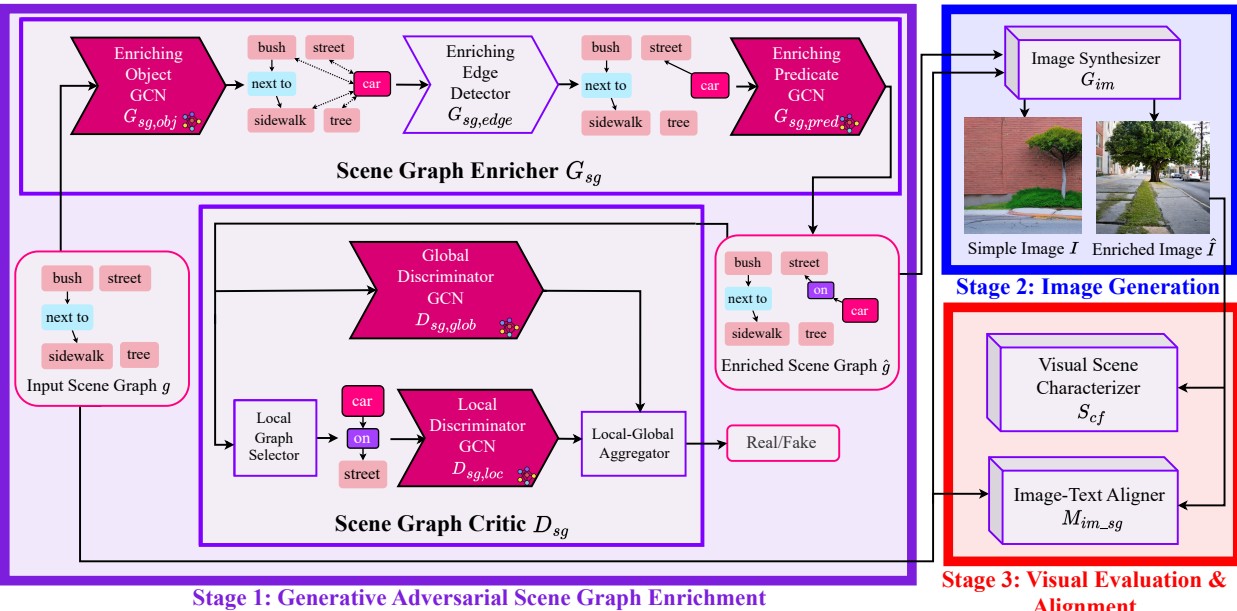

Figure 2: *High-Level Overview* of our end-to-end Generated Contents Enrichment framework during the training phase. In Stage 1, the input scene graph (SG) is fed to the *Scene Graph Enricher $G_{sg}$* to produce an enriched SG. Besides, a pair of local and global discriminators in the *SG Critic $D_{sg}$* are jointly trained to differentiate between original and enriched SGs. These discriminators aid the enrichment process in constructing realistic, structurally coherent, and semantically meaningful scenes. In Stage 2, the *Image Synthesizer $G_{im}$* leverages the resultant enriched SG to generate an image. In Stage 3, essential visual and textual scene characteristics are extracted in the *Visual Scene Characterizer $S_{cf}$* and the *Image-Text Aligner $M_{im\_sg}$*. These two components ensure that the enriched image appropriately reflects the original description's inherent characteristics.

scene graphs from input images, our GCE approach takes a different route by transforming a scene description into an enriched description and generating the corresponding enriched image.

GEMS (Agarwal et al., 2023) expands a seed scene graph for the purpose of image retrieval, iteratively generating novel scenes in the language domain using recurrent neural networks. In contrast, our approach enriches a scene graph entirely within the graph domain, with a clear focus on producing a visually enriched and coherent image. Unlike GEMS, we ensure the enriched scene graph incorporates objects and relationships that are specifically selected to enhance the quality and preserve the coherence of the synthesized image. This distinction is essential as we consider the impact of the enriching elements on the final generated image. In addition, both GCE and GEMS respect object co-occurrence patterns observed in the training data and are capable of generating diverse scene enrichments.

## 3 Method

Fig. 2 illustrates a high-level representation of our end-to-end model during the training phase. The model accepts a scene graph (SG) $g$ as input to simultaneously enrich the content in both textual and visual domains. Our proposed approach for GCE consists of three stages performed iteratively.

Initially, *Stage 1: Generative Adversarial SG Enrichment* accomplishes content enrichment at the object level leveraging GCNs. The *SG Enricher*, functioning as our generator, employs GCNs and Multi-Layer Perceptrons (MLPs) to identify the enriching objects and their inter/intra-relationships. The *SG Critic*, as a pair of local and global discriminators, ensures the enriched parts are realistic, structurally coherent, and semantically meaningful both on their own and within the context of the entire graph that represents the scene as a whole. Eventually, the enriched scene graph advances to *Stage2: Image Generation*, where it serves as the basis for synthesizing an enriched image containing richer relevant content. Lastly, this enriched

image, combined with the input SG, progresses to *Stage 3: Visual Evaluation & Alignment*. Here, we aim to guarantee that the final generated output is visually plausible and faithfully reflects the fundamental scene characteristics outlined in the input prompt. This process involves the employment of a scene classifier and a pair of image/text encoders.

Furthermore, it is important to note that in scenarios where Ground Truth (GT) is unavailable, such as during inference time, both *Stage 3: Visual Evaluation & Alignment* and the *SG Critic* are detached from the network.

### 3.1 Scene Graphs

The input scene graph denoted as $g$ consists of two sets, namely $(O, E)$, accompanied by their corresponding features. Here, $O = \{o_1, \ldots, o_n\}$ signifies the set of objects and $E \subseteq O \times \mathcal{R} \times O$ is the set of edges, where $\mathcal{R}$ is the set of all relationship categories. Each individual object is represented as $o_i \in \mathcal{C}$, wherein $\mathcal{C}$ is the set of all object categories. Additionally, each edge assumes the form of $(o_i, r, o_j)$, where $o_i, o_j \in O$, and the predicate $r \in \mathcal{R}$ describes the relationship type. Within the triplet $(o_i, r, o_j)$, $o_i$ takes on the role of the relationship's subject, $o_j$ functions as the object of the relationship, and $r$ operates as the predicate characterizing the relationship itself. In other words, this formulation essentially employs a directed edge connecting two nodes, where the relationship's nature is encapsulated by the relationship category acting as the predicate.

### 3.2 Graph Convolutional Network (GCN)

Our employed GCN is inspired by Johnson et al. (2018) and produces an output graph that maintains a structure analogous to the input graph. Each GCN is composed of multiple graph convolution (GConv) layers utilized to aggregate the data from nodes and edges, enabling the propagation of information throughout the graph. In this framework, the $l^{th}$ GConv layer, denoted as $GConv^{(l)}$ performs the message-passing as:

$$(V_O^{(l+1)}, V_R^{(l+1)}) = GConv^{(l)}(V_O^{(l)}, V_R^{(l)}, O, E). \tag{1}$$

In the context of the $GConv^{(l)}$'s input graph, $V_O^{(l)}$ represents the feature vectors for all nodes, and $V_R^{(l)}$ denotes the feature vectors for all predicates. Additionally, the corresponding output vectors are characterized as $V_O^{(l+1)}$ and $V_R^{(l+1)}$. To initialize the process, $V_O^{(0)}$ and $V_R^{(0)}$ serve as the initial vector values for the objects and predicates within the GCN input graph.

Like CNNs, our approach employs weight sharing, enabling the GCN to accommodate graphs with an arbitrary number of vertices and edges. Additionally, because the computations for each node are independent, parallelization is achievable, which enhances overall computational efficiency.

The fundamental building block of our GCN is the GConv layer. For every object $o_i \in O$ and edge $(o_i, r, o_j) \in E$, we concatenate the three corresponding sets of feature vectors, which consist of $(v_i, v_r, v_j)$ representing the subject, predicate, and object vectors, respectively. These concatenated feature sets are then passed through the GConv layer for further processing, initiating with an MLP represented as $f_g$, where $(\bar{v}_i, \bar{v}_r, \bar{v}_j) = f_g(v_i, v_r, v_j)$. This operation computes the set of three intermediate subject, predicate, and object vectors $(\bar{v}_i, \bar{v}_r, \bar{v}_j)$. Subsequently, these intermediate vectors are aggregated to form $V_O^{(l)}$ and $V_R^{(l)}$. Node features $V_O^{(l)}$ include the new feature vectors $v_i'$ for the output nodes and edge features $V_R^{(l)}$ consist of the new feature vectors $v_r'$ for the output predicates:

$$v_r' = f_r(v_r, \bar{v}_r), \quad C_i^s = \{\bar{v}_i | (\bar{v}_i, \bar{v}_r, \bar{v}_j) = f_g(v_i, v_r, v_j) \text{ and } (o_i, r, o_j) \in E\}, \tag{2}$$

$$v_i' = f_o(v_i, \frac{1}{|C_i^s| + |C_i^o|} \sum_{\bar{v}_i \in C_i^s \cup C_i^o} \bar{v}_i), \quad C_i^o = \{\bar{v}_i | (\bar{v}_j, \bar{v}_r, \bar{v}_i) = f_g(v_j, v_r, v_i) \text{ and } (o_j, r, o_i) \in E\}. \tag{3}$$

Here, $f_r$ and $f_o$ refer to MLPs that include residual links. Additionally, $C_i^s$ and $C_i^o$ represent candidate vectors for node $o_i$, corresponding to edges where this node serves as either a subject or an object, respectively. To consolidate these candidate vectors for each node, a mean aggregation operation is applied, forming a single

vector $v_i'$. On the other hand, the predicate vector $\bar{v}_r$ is directed through additional neural network layers represented as $f_r$, which generate the new vector $v_r'$ incorporating the effects of the residual link.

For more comprehensive information regarding the structure of our GCN and GConv, including detailed illustrations and further explanations, please refer to the supplementary material.

### 3.3   Stage 1: Generative Adversarial SG Enrichment

### 3.3.1   SG Enricher

The *Scene Graph Enricher*, serving as our SG generator and featured in Fig. 2, encompasses three distinct steps, each tailored to address specific subproblems associated with appending a new object, edges, and predicates. These steps leverage the proposed GCNs along with an *Enriching Edge Detector*. Here is an overview of the process:

First, *Enriching Object GCN $G_{sg,obj}$* takes an input SG $g$ to append an enriching object to the scene. Following the addition of objects in the previous step, *Enriching Edge Detector $G_{sg,edge}$* takes the new graph and the hidden object vectors and appends edges to the scene. Finally, *Enriching Predicate GCN $G_{sg,pred}$* employs the resulting enriched SG from the previous step to detect the predicates associated with the newly added enriching edges. Together, these three steps progressively enhance the input SG by introducing new objects, edges, and predicates, ultimately enriching the scene's content iteratively.

Our *Scene Graph Enricher $G_{sg}$* processes the input graph $g$ as $\hat{g} = G_{sg}(g)$. This operation yields an enriched graph $\hat{g}$. Afterward, $\hat{g}$ is reintroduced into the $G_{sg}$ process, initiating an iterative procedure for further enrichment. This recursive approach allows for the scene's gradual improvement and refinement.

$G_{sg,obj}$ is modeled with the proposed GCN. Additional layers and skip connections are introduced to deepen the GCN, avoiding issues like vanishing gradients. $G_{sg,obj}$ results in the creation of an enriching object denoted as $\hat{o}$, where $\hat{o} = G_{sg,obj}(g)$. This newly generated enriching object is then added to the original graph $g$. The augmented graph, consisting of both the original objects and the newly created object, continues to serve as the input for the succeeding steps of the enrichment process.

$G_{sg,edge}$ is designed with the utilization of two MLPs $\phi_s$ and $\phi_o$ sharing the same architecture but with different sets of weights. These MLPs receive several inputs: the initial input node feature vectors $V_O^{(0)}$, the enriching object $\hat{o}$, and hidden vectors $V_O^{(L)}$ extracted from the final layer of $G_{sg,obj}$. The MLPs $\phi_s$ and $\phi_o$ are specifically designed to capture information from the objects and subjects, respectively, within the context of directed inter-object relationships. This process is clearly illustrated in Eq. 4 and further elucidated in the supplementary material.

$$\hat{M} = \phi_s(V_O^{(0)}, v_{\hat{o}}, V_O^{(L)}) \cdot \phi_o(V_O^{(0)}, v_{\hat{o}}, V_O^{(L)})^T \tag{4}$$

In the *Enriching Edge Detector $G_{sg,edge}$*, each node vector is transformed into another space to determine the location of the new edges within the enriched graph. These transformed object vectors are then compared using the dot product to detect the enriching edges. The similarity of node features in the new space indicates a higher probability of an edge existing between the two corresponding nodes, as reflected in the entries of $\hat{M}$, similar to an adjacency matrix. Consequently, given the relevant entries of $\hat{M}$ related to $\hat{o}$, we select nodes such as $o_e$ that pre-exist in $g$ and form edges $\hat{e} = (o_e, ., \hat{o})$ or $\hat{e} = (\hat{o}, ., o_e)$, where the notation "." represents a placeholder for an unknown predicate. This selection process for nodes forming enriching edges by considering the similarity of node features in the transformed space is summarized as:

$$\hat{e} = G_{sg,edge}(V_O^{(0)}, v_{\hat{o}}, V_O^{(L)}). \tag{5}$$

After the determination of edges between the enriching node $\hat{o}$ and the pre-existing objects $o_e$ is completed, the newly formed graph $\tilde{g} = \{g, \hat{o}, \hat{e}\}$ is then passed to $G_{sg,pred}$. Notably, $G_{sg,pred}$ maintains the same architecture as $G_{sg,obj}$ but operates with an independent set of parameters as

$$\hat{r} = G_{sg,pred}(g, \hat{o}, \hat{e}). \tag{6}$$

$G_{sg,pred}$ is designed with the proposed GCN tasked with producing the type of relationship $\hat{r}$ for each enriching edge $\hat{e}$ derived from the previous step.

Finally, the new graph $\hat{g}$ is assembled by combining the input graph $g$, enriching object $\hat{o}$, the edges $\hat{e}$, and the relationships $\hat{r}$. This comprehensive graph serves as input for the following enrichment iterations and represents the enriched scene with all the appended elements and their associated attributes.

### 3.3.2 SG Critic

*Scene Graph Critic $D_{sg}$*, as depicted in Fig. 2, indeed consists of a pair of discriminators trained jointly with the rest of the network. Some additional details regarding the architecture of GCNs are provided in the supplementary materials. In the global discriminator denoted as $D_{sg,glob}$, a GCN is employed to transform the input into another graph. The extracted features in nodes and edges of this graph are utilized to differentiate the source of input data. Given that our model handles graphs with varying numbers of nodes and edges, an aggregation method such as averaging needs to be utilized at this stage. The output vectors for nodes and edges are separately aggregated and then input into additional neural network layers for further processing and discrimination.

Our global discriminator is deeper compared to the local discriminator represented as $D_{sg,loc}$ by incorporating two additional GConv layers. The local discriminator exclusively receives a subgraph comprising the enriching nodes $\hat{o}$, their immediate neighbors, and the corresponding edges. The outputs of two discriminators are concatenated and subsequently passed through additional layers. This process ultimately culminates in the computation of a score. This score reflects the discrimination of an enriched graph from an original one in our critic's assessment.

### 3.4 Stage 2: Image Generation

Our base image generator employs a pre-trained image synthesizer (Johnson et al., 2018). The process entails feeding scene graphs to a GCN for object features to be extracted. These extracted features are then utilized to form a scene layout for image synthesis with some convolutional layers. Both the GT scene graph $g_{GT}$ corresponding to the input graph $g$ in addition to the resulting enriched scene graph $\hat{g}$ are provided as input to the image generator $G_{im}$ to produce images $I$ and $\hat{I}$ (Eq. 7). These images further contribute to the overall enrichment objective function.

$$I = G_{im}(g_{GT}), \quad \hat{I} = G_{im}(\hat{g}). \tag{7}$$

### 3.5 Stage 3: Visual Evaluation & Alignment

#### 3.5.1 Visual Scene Characterizer

We need to ensure that the enriched output image, with its additional content, faithfully reflects the essential characteristics outlined in the GT description. Thus, a pre-trained scene classifier $S_{cf}$ (Zhou et al., 2017) receives the generated enriched image $\hat{I}$ to evaluate the mentioned criteria. The hidden scene features for both images $I$ and $\hat{I}$ are extracted (Eq. 8). These extracted features are afterward integrated into the objective function, thus playing a crucial role in the end-to-end training of all components within the framework.

$$h_I = S_{cf}(I), \quad h_{\hat{I}} = S_{cf}(\hat{I}). \tag{8}$$

#### 3.5.2 Image-Text Aligner

The *Image-Text Aligner*, represented as $M_{im\_sg}$, leverages pre-trained CLIP encoders (Radford et al., 2021). It aims to ensure that the enriched generated images accurately reflect the essential scene elements outlined in the input description. This multimodal network operates by accepting both the input graph $g$ and the enriched image $\hat{I}$. It extracts features denoted as $F_g$ from the graph and $F_I$ from the image:

$$F_g = M_{im\_sg}(g), \quad F_I = M_{im\_sg}(\hat{I}). \tag{9}$$

These features are further employed in the objective function.

### 3.6 Objective Function

To enable end-to-end training, we randomly eliminate one object from each GT scene graph $g_{GT}$, along with its associated edges. The modified graph is then used as the input scene graph $g$. The goal of $G_{sg,obj}$ is to predict the object category of this removed node, which leads to our *Obj. Loss*,

$$\mathcal{L}_{obj} = \frac{1}{N} \sum_{i=1}^{N} CE(\hat{o}^{(i)}, o_{GT}^{(i)}), \tag{10}$$

where $N$ represents the mini-batch size. This loss is structured as a cross-entropy measure, comparing the predicted object $\hat{o}^{(i)}$ and its GT counterpart $o_{GT}^{(i)}$ in the GT graph $g_{GT}$. In fact, the eliminated object is denoted using an introduced distinct object category labeled as *unknown_obj*. Additionally, all the potential edges leading to or from this node are included as *unknown_pred*, a newly introduced predicate category.

$G_{sg,edge}$ produces a matrix $\hat{M}$ which resembles an adjacency matrix. However, each entry $\hat{m}_{i,j}$ within this matrix signifies the probability of an edge existing between $o_i$ as a subject and $o_j$ as an object. The *Enriching Edge Detector* contributes to the objective function with *Edge Loss*,

$$\mathcal{L}_{edge} = \frac{1}{N} \sum_{k=1}^{N} \sum_{i,j} BCE(\hat{m}_{i,j}^{(k)}, m_{GT;i,j}^{(k)}) \tag{11}$$

computed as a binary cross-entropy considering whether an edge exists between each pair of nodes in the GT graph, including the enriching object.

Based on $\hat{M}$, one of the edges could be selected, and then its predicate is predicted by $G_{sg,pred}$. In the GT, the predicate for this edge may or may not be provided, leading to the formulation of two supplementary loss terms computed via cross-entropy as *Available Preds Loss* and *Not Avail. Pred. Loss*:

$$\mathcal{L}_{pr,a} = \frac{1}{N} \sum_{i=1}^{N} CE(\hat{r}^{(i)}, r_{GT}^{(i)}), \quad \mathcal{L}_{pr,na} = \frac{1}{N} \sum_{i=1}^{N} CE(\hat{r}^{(i)}, none\_pred), \tag{12}$$

where the term *none_pred* signifies a predicate category analogous to *no relationships*.

We extract a pair of features from the final layer of the scene classifier, one corresponding to the generated image from the GT scene graph and the other to the enriched synthesized image (Eq. 8). These features are then compared, leading to the alignment process contributing to enrichment. This alignment is to ensure that the appended content is coherent with the high-level essential visual characteristics of the GT. This loss term is denoted by *Scene Classifier Loss*,

$$\mathcal{L}_{sc} = \frac{1}{N} \sum_{i=1}^{N} ||h_I^{(i)} - h_{\hat{I}}^{(i)}||_p, \tag{13}$$

implemented as $L_p$ difference between GT and enriched scene features. Alternatively, it can be modeled as another form of comparison:

$$\mathcal{L}_{sc} = \frac{1}{N} \sum_{i=1}^{N} CE(s_{\hat{I}}^{(i)}, s_I^{(i)}), \tag{14}$$

the cross-entropy between the predicted scene classes denoted as $s_I$ for image $I$ and $s_{\hat{I}}$ for image $\hat{I}$. This constraint guarantees that the predicted scene classes are identical, thereby signifying that the enrichment content does not modify the high-level scene comprehension of the input description.

The *Image-Text Aligner* produces a pair of normalized features, denoted as $F_g$ and $F_I$ both of which share the same dimension (Eq. 15). These features are then used to establish the *Image-SG Alignment Loss*,

$$\mathcal{L}_{im\_sg} = -\frac{1}{N} \sum_{i=1}^{N} F_g^{(i)} \cdot F_I^{(i)}, \tag{15}$$

which takes the form of a cosine similarity. This similarity measure is applied to maintain the consistency of the synthesized enriched image in relation to the fundamentals outlined in the original description.

$D_{sg}$ and $G_{sg}$ undergo training within an adversarial setting by engaging in a GAN min-max game. In this setting, the generator and the discriminators have opposing objectives, seeking to minimize and maximize the following loss term, respectively:

$$\mathcal{L}_{GAN} = \underset{g \sim p_{\text{real}}}{\mathbb{E}} \log D_{sg}(g) + \underset{\hat{g} \sim p_{\text{enriched}}}{\mathbb{E}} \log(1 - D_{sg}(\hat{g})) \tag{16}$$

Finally, in our end-to-end training process, we opt for a weighted sum $\mathcal{L}$ that encompasses all the previously mentioned loss terms (Eq. 17). This comprehensive objective function promotes the training of the entire framework, ensuring that all components work together cohesively.

$$\mathcal{L} = w_0 \mathcal{L}_{obj} + w_1 \mathcal{L}_{edge} + w_2 \mathcal{L}_{GAN} + w_3 \mathcal{L}_{pr,a} + w_4 \mathcal{L}_{pr,na} + w_5 \mathcal{L}_{sc} + w_6 \mathcal{L}_{im\_sg} \tag{17}$$

A thorough exploration of training specifics and hyperparameter selection can be found in the supplementary materials. This encompasses various configurations for our GCNs' architectures, mini-batch size, activation functions, dimensions of embedding vectors within the generator and discriminators, dropout layers, normalization techniques, and different combinations of weights for loss terms as defined in Eq. 17.

During the training process, gradients are backpropagated through the pre-trained versions of the image generator, scene classifier, and CLIP encoders. However, it is essential to note that only the parameters associated with $G_{sg}$ and $D_{sg}$ are updated as part of the training procedure.

During the inference phase, an *unknown_obj* node along with its corresponding *unknown_pred* edges are introduced into the input graph before being passed to $G_{sg}$. Among the *unknown_pred* edges from the *unknown_obj* to all the other objects, we preserve only the edge with the highest probability of existence, as predicted by $G_{sg,edge}$. The enrichment process can be iteratively performed multiple times to progressively add additional content. At each iteration of this process, one object and one predicate are added to the description, enabling the generation of images from the resulting enriched intermediate scene graphs.

## 4 Experiments

In this section, we aim to establish the coherence between the enriching objects and relationships with the original scene as described by the input scene graph. Additionally, we demonstrate that scene graph enrichment leads to an augmented version of the input graph, ultimately generating richer images. We achieve these through a combination of qualitative illustrations and quantitative results.

Given the absence of existing work on graphs that directly addresses the specific task we tackle, our primary focus is to demonstrate the promise of the proposed approach. We also include a comparison with ChatGPT as a Large Language Model (LLM). Our objective is to illustrate the feasibility of enriching generated content rather than surpassing SOTA text/graph-to-image models, GNNs, or scene classifiers. Other end-to-end trainable modules with similar functionality can be integrated into our approach to achieve potentially enhanced performance.

During the inference phase, we utilize Stable Diffusion (Rombach et al., 2022), a more advanced image synthesizer compared to SG2IM (Johnson et al., 2018), which is employed during the training phase. This allows for enhanced visualization of details without the computational constraints imposed during training.

### 4.1 Dataset

We utilized the Visual Genome (VG) dataset (Krishna et al., 2017), which consists of 110k real-world images with their scene graphs. We employed a preprocessing similar to Johnson et al. (2018). This resulted in 178 object and 45 predicate categories in approximately 62.5k, 5.5k, and 5k samples for training, testing, and validation, respectively. VG offers rich and detailed annotations, comprising 3.8M objects and 2.3M relationships, which capture a diverse range of environments and object configurations

The COCO-Stuff dataset (Caesar et al., 2018) with only 25k training images, is often used alongside VG (Johnson et al., 2018; Yang et al., 2022c; Herzig et al., 2020). Like VG, COCO-Stuff selected its images from the MS COCO dataset (Lin et al., 2014). However, COCO-Stuff lacks scene graphs and only provides bounding boxes. Thus, scene graphs are synthesized based on the 2D coordinates of objects, with only six automatically generated relationships: left of, right of, above, below, inside, and surrounding.

Due to the synthetic nature and limited relational scope of COCO-Stuff, we believe it is unsuitable for demonstrating the full potential of our model's enrichment capabilities, as the generated graphs do not accurately reflect real-world descriptions. To the best of our knowledge, no other non-synthetic scene graph datasets are available for our purposes. In contrast, the VG dataset's detailed and realistic annotations enable our model to generalize effectively, generating a wide variety of scenes with complex and diverse object relationships.

## 4.2 ChatGPT Comparison Experiments

In case the input is represented as raw text rather than modeling as scene graphs, LLMs may be employed to expand the scene descriptions (Brown et al., 2020; OpenAI, 2023; Touvron et al., 2023; Scao et al., 2022; Ouyang et al., 2022; Anil et al., 2023; Driess et al., 2023). However, LLMs cannot efficiently and effectively perform the GCE task. First, simultaneous training of LLMs and image generators demands substantial training data and entails significant computational expenses. Secondly, despite the LLMs' capacity to enrich the textual information, they lack the ability to comprehend the visual implications of the augmented content and measure its adequacy for generating a single real-world image. Thirdly, the direct utilization of LLMs might result in redundant content generation for image synthesis, particularly given that even the SOTA generation method (Rombach et al., 2022) struggles with handling exceedingly complex prompts.

To illustrate the superiority of our proposed framework over LLMs, we conducted a comparison utilizing ChatGPT (OpenAI, 2023) for the content enrichment task. This is presented qualitatively in Fig. 3 and quantitatively in Table 2. The input descriptions are provided to ChatGPT, which generates enriched descriptions that are then used to synthesize images.

We limited the word count for ChatGPT output as originally it generates very long texts unsuitable for inputs to SOTA image generators. These three different styles of prompts were used: (i) *ChatGPT Direct:* Please complete this given text so that the final text contains between 15 to 25 words: "*input_description*". (ii) *ChatGPT Scene:* Please complete this given text **describing a scene** so that the final text contains between 15 to 25 words: "*input_description*". (iii) *ChatGPT Object:* Please complete this given text **describing a scene with more related objects** so that the final text contains between 15 to 25 words: "*input_description*".

## 4.3 Qualitative Results

Fig. 3 showcases input descriptions and their enrichments alongside their corresponding synthesized images employing *ChatGPT Direct* and our proposed model. These input scene graphs are selected from the VG test split. The input scene graphs, along with the enriched scene graphs generated by our model, are represented as textual descriptions in Fig. 3 (a) and (c).

In cases where the original scene graph contains a large number of objects, a subgraph with its associated edges is randomly selected and used as the initial input for our experiments. This subgraph strategy is primarily driven by the limitations of image generators when synthesizing images with lots of objects. Thus, if the initial input description already contains a substantial number of objects, the image synthesizer cannot accurately depict the enriched scene with even more objects. However, it is essential to note that there are no constraints regarding the number of semantics in the input description. Our proposed method carries out semantic enrichment iteratively, focusing on enriching one object at a time in each step, regardless of the number of existing objects in the scene.

To ensure there is no overlap in VG between the data used for training different stages of our model, *Stage1: Image Generation* is pre-trained on the same train split as ours. Moreover, in *Visual Scene Characterizer*, we employ a pre-trained residual CNN scene classifier (Zhou et al., 2017). During network training, the

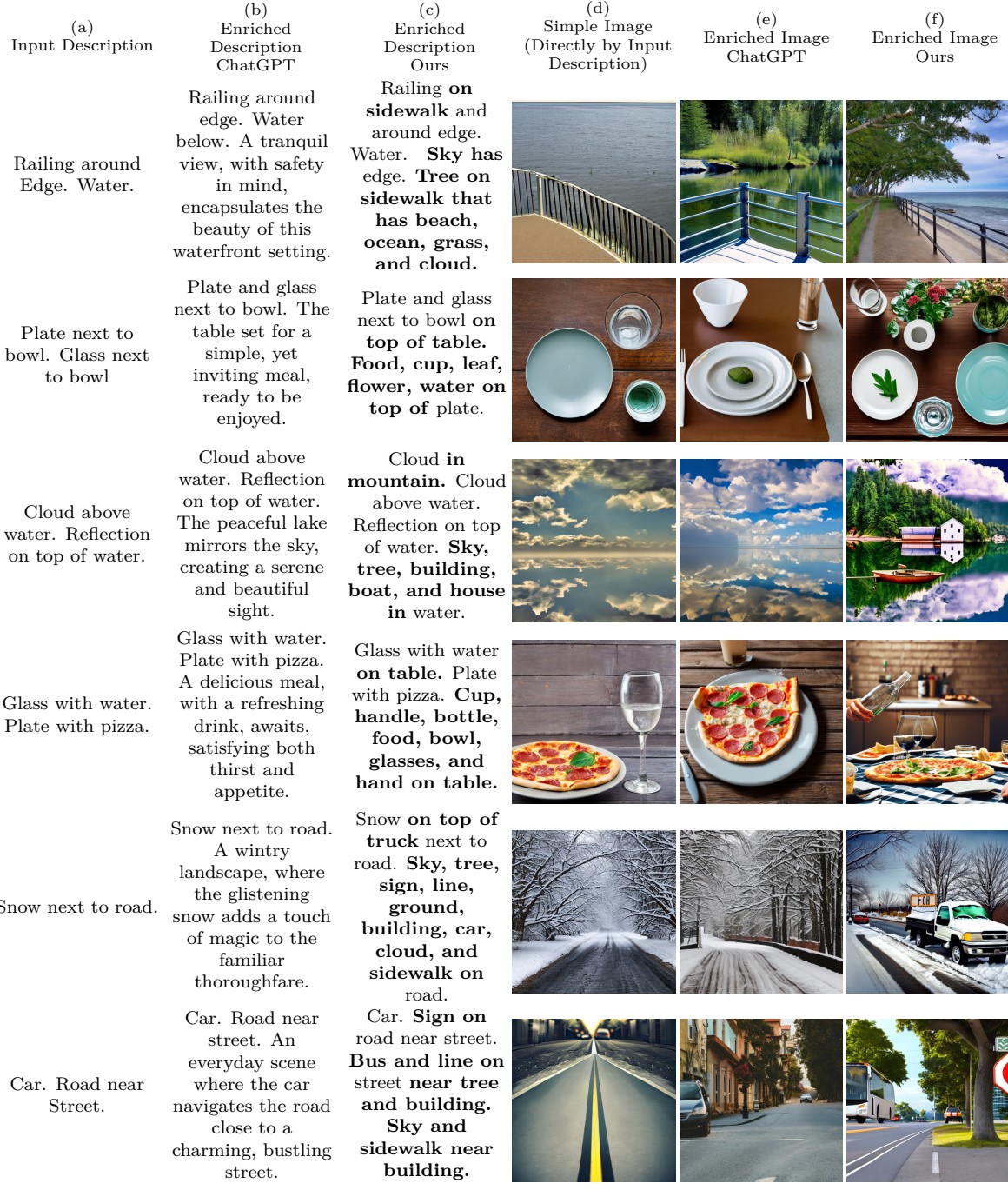

Figure 3: *Qualitative Results.* Samples of scene graphs from the Visual Genome test split as the input descriptions are enriched, along with their synthesized images featuring richer content. The simple image is generated directly from the input description, and the enriched images are produced from their corresponding enriched descriptions. The input scene graphs (a) and the enriched scene graphs generated by our model (c) are represented as textual descriptions.

Table 1: Our defined measures for prediction accuracy are presented across selected hyperparameter configurations. The supplementary materials provide details on the models. The enrichment task is inherently ill-posed, allowing for multiple correct solutions. However, our method enriches content in a structurally reasonable manner, avoiding random generation.

| Measure | SG Enrich | Deep G | Best NoD | 128 BS |
|---|---|---|---|---|
| Obj. Acc. | 19.04 | 17.31 | 15.09 | 16.85 |
| Avail. Pred. Acc. | 41.95 | 39.90 | 40.25 | 31.38 |
| Not Avail. Pred. Acc. | - | - | 3.02 | - |
| Avail. Edge Acc. | 75.00 | 75.64 | 76.08 | 66.89 |
| Not Avail. Edge Acc. | 99.58 | 99.70 | 99.54 | 99.54 |
| Scene Class. Acc. | 20.33 | 20.84 | 19.18 | 25.33 |

gradients backpropagate through *Stage 3: Visual Evaluation & Alignment* and *Stage2: Image Generation*; however, their parameters are not updated, as our objective function is specifically designed to optimize a different problem which is joint SG/image enrichment.

### 4.4 Quantitative Results

Evaluating the performance of SG enrichment presents a non-trivial challenge, given that there is no unique solution. In other words, there are multiple ways to enrich a graph by adding objects and relationships, each leading to a richer generated image with reasonable semantics.

### 4.4.1 Inpainting Inspired Metrics

Similar to the domain of image inpainting, we may seek simple measures comparing predictions and GT, such as $L_p$ distance for pixel values or, in our specific case, accuracy for object and relationship predictions.

As many terms are encompassed within our loss function, the contribution of each term plays an essential role in the overall enrichment task. For this reason, we conducted comprehensive hyperparameter tuning. This includes the selected hyperparameters corresponding to the models in Table 1, providing our quantitative evaluation methods for some of the chosen instances. *SG Enrich* denotes our proposed method with optimized hyperparameters. *Deep G* refers to a version utilizing a deeper generator architecture. *Best NoD* represents the optimal scenario without the introduced discriminator, while *128 BS* signifies the use of a significantly larger mini-batch size. Detailed results and discussions can be found in the supplementary materials. Table 1 demonstrates that our proposed method is capable of content enrichment in a structurally reasonable manner, as opposed to merely adding random content. It is important to emphasize that the enrichment process is inherently ill-posed since any appropriate enrichment is correct, and the final result is not deterministic. Therefore, the relatively low prediction accuracy does not provide a comprehensive illustration of the real performance of our introduced approach.

The metric *Obj. Acc.* represents the accuracy of our prediction among approximately 178 object categories. While it may appear relatively low, it is important to note that this outcome arises from the nature of our task. We randomly eliminate an object from a scene and then attempt to predict it based on the semantics and structure of the scene. Moreover, the GT scene graph is not the only reasonable answer. Thus, different objects, other than the GT, may be appended to form visually plausible enriched images while maintaining the reasonable structure and coherence of the entire scene.

The edge detection task involves binary predictions between every pair of nodes. During the training of the *Enriching Edge Detector*, predictions are made for all objects in the scene graph, encompassing both the enriching and original objects.

*Avail. Edge Acc.* measures the accuracy of predicting the presence of a directed edge between two nodes connected as subject and object in a relationship within the GT graph. Similarly, *Not Avail. Edge Acc.* represents the accuracy for edges whose existence is not provided in the GT graph. It is worth mentioning that the absence of an edge in the GT graph does not necessarily imply that predicting such an edge would lead to a description of a different scene. Therefore, any comparisons between the results should be made cautiously, as the problem inherently allows for multiple valid solutions.

Table 2: Inception Scores (IS) (Salimans et al., 2016) and Fréchet Inception Distances (FID) (Heusel et al., 2017) are computed to compare real-world **VG** dataset and sets of images generated from GT scene descriptions (**GT Synthesized**), **Simple** descriptions, and **Enriched** versions utilizing **Our** model and **ChatGPT** with three different prompt styles.

| Metric [Reference Distribution] | Simple | Enriched ChatGPT Direct | Enriched ChatGPT Scene | Enriched ChatGPT Object | Enriched Ours | GT Synthesized | VG | Improvement Ours vs. Simple |
|---|---|---|---|---|---|---|---|---|
| IS↑ | $17.03 \pm 0.37$ | $15.44 \pm 0.32$ | $14.90 \pm 0.19$ | $15.69 \pm 0.26$ | $\mathbf{19.37 \pm 0.28}$ | $24.44 \pm 0.32$ | $38.61 \pm 0.52$ | 13.74% |
| FID↓ [GT Synthesized] | 37.85 | 42.15 | 47.74 | 52.74 | **34.89** | 0 | 35.32 | 7.82% |
| FID↓ [VG] | 61.27 | 64.69 | 68.06 | 73.48 | **53.23** | 35.32 | 0 | 13.12% |

In cases where the *Enriching Edge Detector* fails to accurately predict the existence of an enriching edge compared to the GT, we also penalize the *Enriching Predicate GCN*. This is achieved using the appropriate loss term and the introduced *none_pred* as the GT predicate category. The accuracy of this prediction is denoted as *Not Avail. Pred. Acc.*, which is included in Table 1 only if the edge prediction fails for at least one enriching object, as per the GT graph. However, assume the *Enriching Edge Detector* successfully performs its task based on the GT. Then, we consider the prediction of the *Enriching Predicate GCN* as *Avail. Pred. Acc.*. This metric assesses the accuracy of predicting the predicates among approximately 45 different relationship categories.

*Scene Class. Acc.* demonstrates whether the scene categories for a pair of images generated from GT and enriched scene graph are the same. During training, images have a lower resolution, and the pre-trained scene classifier does not have a high classification accuracy on its own. Thus, we do not anticipate a high matching accuracy for *Scene Class. Acc.*. Though the GT scene category is not unique, our goal is to enrich a scene by appending only one object and its relationships to finally synthesize images with the same essential visual scene characteristics as the GT graph.

### 4.4.2 Generated Image Quality Metrics

Inception Scores (IS) (Salimans et al., 2016) and Fréchet Inception Distances (FID) (Heusel et al., 2017) are widely recognized metrics used to evaluate the quality of synthesized images. IS relies on a pre-trained image classifier to assess the presence of recognizable objects and their diversity within the generated image set. FID, on the other hand, employs a pre-trained classifier to extract deep features from both generated and real image sets. It then measures the statistical similarity between the distributions of these extracted features for the generated and real image sets.

Table 2 exhibits FID and IS metrics for evaluating the image quality. Stable Diffusion is employed as the image synthesizer to generate images for different scenarios compared to the real-world VG images: (i) First, simple scene descriptions from the test set are utilized for synthesizing a total of 36.5k simple images denoted by *Simple*. These represent synthesized images produced by a SOTA image generator (Rombach et al., 2022) without incorporating any explicit content enrichment model. (ii) Second, the same simple descriptions are fed to our proposed GCE method or ChatGPT with the three introduced prompting styles. This is to produce the enriched descriptions used for synthesizing 36.5k enriched images denoted by *Enriched* for each of the mentioned four cases: *Enriched ChatGPT Direct/Scene/Object* and *Enriched Ours*. (iii) Third, GT training scene descriptions are directly used to generate 35k images denoted by *GT Synthesized*. (iv) The fourth set of images is 87k real-world VG images denoted by *VG*.

The FID metric is calculated with two different reference distributions denoted by *FID [GT Synthesized]* and *FID [VG]*: one considering *GT Synthesized* images and the other one using *VG* images as reference distributions, respectively. The results show improvements in both IS and FID metrics for *Enriched* images compared to *Simple* ones. This coarsely demonstrates that our enriched images have higher-quality content and similarity to the reference distributions. However, it is important to note that *GT Synthesized* images maintain better IS and FID metrics than *Enriched* images. This is because *GT Synthesized* images are generated directly from the GT training split, which provides more detailed and diverse descriptions. Thus, when using a fixed image synthesizer, *GT Synthesized* may be considered an upper bound for the enrichment task.

Due to limitations in the VG dataset, our model does not enrich the attributed properties of each object, such as its color. However, these attributes are present in ChatGPT-enriched texts, which adds diversity to

Table 3: *Ablation study results.* Modules are altered with respect to *SG Enrich* as the baseline.

| Measure | SG Enrich | W/o $D_{sg}$ | W/o $D_{sg,glob}$ | W/o $D_{sg,loc}$ | W/o $S_{cf}$ | With $M_{im\_sg}$ |
|---|---|---|---|---|---|---|
| Obj. Acc. | 19.04 | 18.33 | 18.52 | 18.66 | 17.88 | 19.48 |
| Avail. Pred. Acc. | 41.95 | 37.43 | 38.87 | 37.47 | 36.08 | 42.20 |
| Not Avail. Pred. Acc. | - | - | - | - | - | - |
| Avail. Edge Acc. | 75.00 | 74.89 | 74.20 | 75.33 | 74.62 | 75.18 |
| Not Avail. Edge Acc. | 99.58 | 99.61 | 99.61 | 99.60 | 99.66 | 99.57 |
| Scene Class. Acc. | 20.33 | 19.99 | 21.14 | 21.66 | - | 22.66 |

the generated images, potentially improving IS and FID. As a result, ChatGPT might be placed in a superior prior position compared to our proposed model. Nonetheless, when evaluating FID and IS, our introduced framework outperforms ChatGPT, which performs worse than even *Simple* initial descriptions. Notably, as we include specific information related to the task, transitioning from *ChatGPT Direct* to *Scene* and then *Object*, FID scores worsen, indicating that LLMs such as ChatGPT, in their standard out-of-the-box version, are not well-suited for the GCE task. This is primarily due to LLMs generating redundant content for image synthesis without considering the visual implications of the enriched text for producing a realistic image.

## 4.5 Ablation Study

In order to demonstrate the necessity of each component in our model and their individual contribution to the overall performance, we conducted an ablation study. The results of this study are presented in Table 3, where the ablated models are compared employing our defined metrics for scene graph enrichment. Since this task lacks a unique solution, raw results with the specified metrics may be misleading for comparison and often can contradict human judgments.

The baseline for these studies is the *SG Enrich* model with hyperparameters and architectures detailed in the supplementary materials. We explore various cases, including eliminating the scene classifier and adding the *Image-Text Aligner*. The pair of discriminators or each of the local and global discriminators individually is also removed to coarsely indicate their contribution.

In the case labeled **W/o $D_{sg}$**, we omit the pair of discriminators, effectively eliminating the adversarial GAN loss component, which backpropagates from the discriminators to aid the enrichment process.

**W/o $D_{sg,loc}$** and **W/o $D_{sg,glob}$** are cases with only global or local discriminators retained, respectively. $W/o$ $D_{sg,loc}$ understands the entire scene without paying specific attention to the enrichments. Although the individual enriched subgraphs may seem realistic in $W/o$ $D_{sg,glob}$, it is incapable of capturing the overall inconsistency between the enriched content and the entire scene.

**W/o $S_{cf}$** omits the pre-trained scene classifier in the *Visual Scene Characterizer*. This component evaluates whether the synthesized images reflect the essential scene characteristics described in the original input. Though some objects in the synthesized images might often not be recognizable for the scene classifier, the related loss term still contributes to the overall performance, helping the model to enrich scenes effectively.

The variant labeled **With $M_{im\_sg}$** includes the *Image-Text Aligner* component, which is not present in our baseline model denoted by *SG Enrich*. This component comprises the CLIP encoders accountable for ensuring that the generated enriched images accurately reflect the essential scene components described in the original input scene graphs.

## 4.6 User Studies

Our efforts to demonstrate the performance of our model and its different variations involved employing our defined metrics and borrowed measures from the assessment of generated image quality. Nevertheless, it is crucial to acknowledge that evaluating the model's performance solely against GT provides only a rough measure due to the inherently ill-posed nature of the problem. Ultimately, human judgment is essential to truly claim the quality of the enrichment process. In this regard, our user studies, detailed in Table 4, were conducted with the participation of 110 individuals, spanning four sections.

Table 4: User studies with 110 participants are conducted to assess the performance and realism of generated images. S1 shows that users can generally distinguish AI-generated images, probably due to their vivid appearance. S2 highlights that our enriched images are perceived as realistic and structurally plausible. S3 reveals our model generates enriched content while reflecting the original description. S4 confirms user preference for our enriched images due to their coherence and meaningful content.

| **S1:** Which one of the images is more realistic? | | |
|---|---|---|
| **(A) Stable Diffusion Image** | **(B) Real-World Image** | **± Standard Deviation** |
| 20.09% | 79.91% | ± 23.27% |
| **S2:** Which one of the images is more realistic? | | |
| **(A) Enriched Image** | **(B) Simple Image** | **± Standard Deviation** |
| 49.45% | 50.55% | ± 28.30% |
| **S3:** Does the image reflect the description? | | |
| **(A) Yes** | **(B) No** | **± Standard Deviation** |
| 70.82% | 29.18% | ± 13.92% |
| **S4:** Based on real image, which AI-generated image do you prefer? | | |
| **(A) Enriched Image** | **(B) Simple Image** | **± Standard Deviation** |
| 83.40% | 16.60% | ± 12.24% |

**S1: Real vs. Stable Diffusion Images Realisticness.** To showcase the capability of the image generator to produce *realistic* images, we presented users with ten pairs of images. Each pair consists of an image selected from either the VG or MS COCO (Lin et al., 2014) and an image generated by Stable Diffusion, both depicting similar scenes. Participants were asked: *Which one is more realistic?* Surprisingly, only in 20.09% of cases did participants select the synthesized images as more realistic. In most instances, users tend to successfully identify the AI-generated images, probably attributed to the vivid and visually appealing nature of Stable Diffusion images, which imitate the characteristics of their training data.

**S2: Simple vs. Enriched Images Realisticness.** For ten simple descriptions extracted from subgraphs in the test split and their corresponding enriched descriptions produced by our model, pairs of images are synthesized. Users were asked: *Which one is more realistic?* The enriched images contain more objects and richer content, increasing the potential for artifacts, while the simple images represent minimalistic everyday scenes. However, participants found the enriched images seem as realistic as the simple ones. In essence, they were unable to distinguish between the two in terms of realisticness. This study indicates that the scenes depicted in our enriched images maintain coherence and structural plausibility, incorporating semantics that do not construct unrealistic scenes.

**S3: Simple Description & Enriched Images.** Ten pairs of simple descriptions, analogous to those in S2, were presented to participants alongside their corresponding enriched images. The question posed to the participants was: *Does the generated image reflect the description?* This investigation illustrates that our model can produce enriched versions of descriptions and images with richer content while faithfully reflecting the principal components of the original input description.

**S4: Simple vs. Enriched Images Preference.** In this evaluation, 14 triplets of images were displayed to the users. Each triplet includes a pair of simple and enriched images, similar to S2, alongside their real-world VG images corresponding to identical scene descriptions. Users were explicitly informed which two images were AI-generated while specifying the real one. They were asked: *Based on the real image, which AI-generated image do you prefer?* The outcomes of this study reveal that users prefer our enriched images, which offer meaningful content and adhere to structural coherence.

## 5 Conclusion

In this paper, we addressed the task of *Generated Contents Enrichment* by proposing an innovative approach. A novel adversarial GCN-based architecture accompanied by its affiliated multi-term objective function is introduced. We synthesize an enriched version of the input description, resulting in richer content in both visual and textual domains. This method aims to tackle the semantic richness gap between human hallucination and the SOTA image synthesizers. We guarantee that the final enriched image accurately reflects the essential scene elements specified in the input description. Accordingly, a pair of discriminators and alignment modules ensure that the explicitly appended objects and relationships cohesively form a structurally and semantically adequate scene.

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

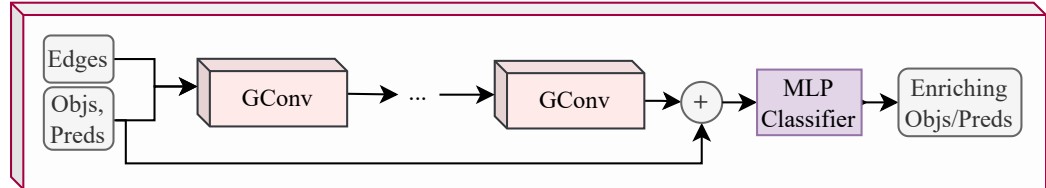

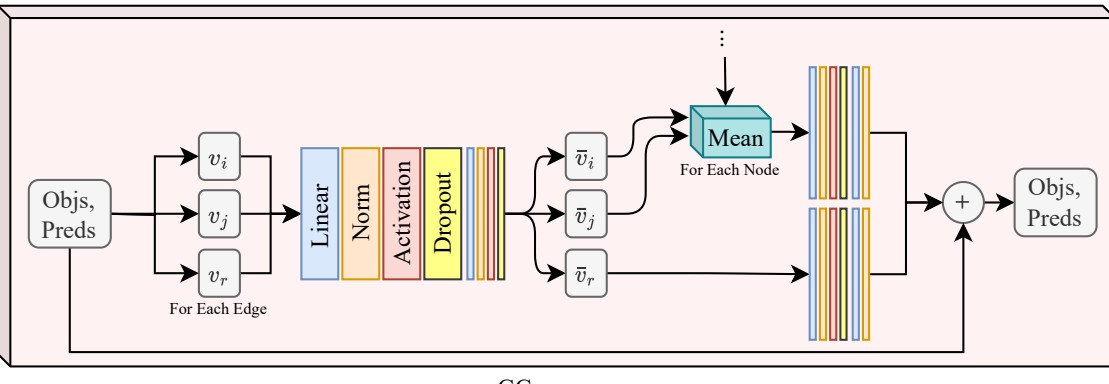

Figure 4: *Graph Convolutional Network (GCN) and its building block GConv are employed in Stage 1: Generative Adversarial SG Enrichment.*

## A Method

### A.1 GConv Operation

Our GCN and its building block GConv are depicted in Fig. 4. Elements of an edge are $(o_i, r, o_j)$ with corresponding vector values $(v_i, v_r, v_j)$ and dimensions $(D_{in}, D_{in}, D_{in})$. They are concatenated and then fed into a graph convolution layer. At the start of the GConv operation, there are two fully connected (FC) layers with output dimensions $H = 2(D_{in} + D_{out})$ and $3H$. The output of these FC layers is then divided into three parts, which represent candidate vectors for the subject node, the edge, and the object node. They are represented as $(\bar{v}_i, \bar{v}_r, \bar{v}_j)$ with dimensions $(H, H, H)$. $\bar{v}_r$ is then fed into additional layers for the skip connection. If this operation is performed for every edge, in the end, for each node, we may have some candidate object vectors and some other candidate subject vectors. By computing the mean of these candidate vectors for each node and then feeding into additional FC layers with output dimensions $H$ and $D_{out}$, a new vector is calculated for each node after the skip connection. A new graph could be formed with the new vectors for the edges and nodes, which is then fed into the following graph convolution layers. There is also an additional skip connection link from the input graph to the final layers of the GCN. This is because we may train deeper networks using skip connections without gradient vanishing problems.

### A.2 GCN Implementation

For processing a mini-batch in parallel, all the edges in the mini-batch graphs are fed into a graph convolution layer simultaneously. Similar to Johnson et al. (2018), in order to make each SG separate from others, it is made connected using a dummy node with *image* as its category. Each node of the SG is connected to this dummy node with directed edges and *in_image* as their categories. In addition, two other vectors corresponding to objects and edges showing the corresponding SG placement in the mini-batch are fed to the networks.

### A.3 Generator's GCNs Architecture

The architecture of the generator's two GCNs is the same but with different weights. First, the dimension of the data is increased to form a subspace where we can do the processing, then reduced back to a low

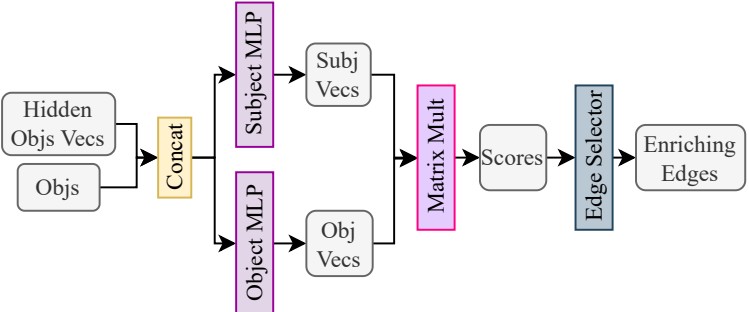

Figure 5: *Enriching Edge Detector* receives the nodes of a graph but not their edges. It also accepts hidden object vectors from the *Enriching Object GCN* as input. After concatenation, each node is fed to two neural networks with the same architecture but separate sets of weights to form hidden vectors for subjects and objects of a relationship. This part aims to transfer the nodes to another space where cosine similarity represents the potential of existing edges between two nodes. Therefore, the resulting subject and object vectors form a score matrix implemented as matrix multiplication.

dimension to encourage the network to extract an encoded version of input information. Finally, it returns to its original dimension. The number of neural network layers in the GCN classifiers and *Enriching Edge Detector* model are selected via *GCN Classifier Layers* and *Enriching Edge Layers* in Table 5. In addition, at the first layers of our GCNs, two embedding linear layers without bias work as object and predicate encoders to transform the one-hot-encoded categories of objects and relationships into another continuous space.

### A.4 CLIP Text Encoder

As the CLIP text encoder cannot handle the scene graphs directly, each edge is converted to a sentence, including the subject, the predicate, and the object of the relationship. These sentences are fed to the CLIP text encoder in addition to the objects that do not participate in any relationship as a text prompt.

## B Experiments

### B.1 Image Generator

In some cases, scenes and objects synthesized with SG2IM (Johnson et al., 2018) are unrecognizable even for humans, which impacts the understanding of the scene by the *Stage3: Visual Evaluation & Alignment*. We attempted to utilize other SOTA image generators instead of SG2IM during training, such as Stable Diffusion (Rombach et al., 2022), which produces $512 \times 512$ and $768 \times 768$ images in the most recent version. However, because of the iterative nature of their inference, generating every single image of the mini-batch without considering the backpropagation is computationally expensive. Therefore, it is impossible to get any reasonable results while doing the hyperparameter tuning in a timely manner employing more sophisticated diffusion-based high-resolution image generators.

### B.2 Pair of Discriminators

On the VG dataset with the selected architectures, distinguishing the enriched SGs from the original data is usually far more manageable than the enrichment task. That is why our discriminators often perform far better than the generator. We tried some instances to limit the discriminator compared to the generator counterpart by having an embedded dimension selected from 16 to 256, smaller than the generator's embedded dimension. Updating the discriminators after each generator update makes them far more powerful than their counterpart generator. That is why different settings for updating the discriminators only one time for several updates of the generator are utilized based on *Update Every* in Table 5.

Table 5: *Hyperparameter Selections* for training the network. As several loss terms are involved in the training process and considering that different values for their weights significantly impact the overall performance, we did a sweep over these options. Also, for training speed and generalizability, we tried out different options for dropout, normalization, and architecture settings. Besides, GConv HP Category is applied in both the generator and the discriminators. Also, the embedded dimension determines the data dimension after feeding the vectors to the embedding layers. In addition, the underlined values are eliminated using preliminary training only on the *Generative Adversarial Scene Graph Enrichment* module.

| HP Category | HP | Values |
|---|---|---|
| Batch | Size | 32, 64, 128 |
| GConv (D, G) | Dropout | 0, 0.05, 0.1, 0.15, 0.25, 0.5 |
| | Normalization | none, batch, layer |
| | Activation | ReLU, LeakyReLU, PReLU |
| Loss | Obj. Weight | 0.001, 0.01, 0.1, 0.5, 1, 5, 10, 50, 100, 200, 500, 1000 |
| | Avail. Pred Weight | 0.001, 0.01, 0.1, 0.5, 1, 5, 10, 50, 100 |
| | Not Avail. Pred Weight | 0, 0.001, 0.01, 0.1, 0.5, 1, 5, 10, 50, 100 |
| | Edge Weight | 0.001, 0.01, 0.1, 0.5, 1, 5, 10, 50, 100 |
| | GAN Weight | 0, 0.001, 0.01, 0.1, 0.5, 1, 5, 10 |
| | Scene Features Weight | 0.001, 0.01, 0.1, 0.5, 1, 5, 10, 50, 100, 200, 500, 1000 |
| | Scene Features | $\text{logit}_{l_1,l_2,CE}$, $\text{hidden}_{l_1,l_2}$, $\text{hpooled}_{l_1,l_2}$ |
| | Image-SG Alignment Weight | 1, 10, 100, 1000, 2000, 5000, 10000 |
| Generator | Embedded Dim. | 16, 32, 64, 128, 256 |
| | GCN Architecture | (a) 1 1 1 1 1 |
| | | (b) 1 1 1 1 1 1 1 |
| | | (c) 1 4 2 2 4 1 |
| | | (d) 1 4 2 1 1 2 4 1 |
| | | (e) 1 4 2 1 $\frac{1}{2}$ $\frac{1}{2}$ 1 2 4 1 |
| | | (f) 1 4 2 1 $\frac{1}{2}$ $\frac{1}{4}$ $\frac{1}{4}$ $\frac{1}{2}$ 1 2 4 1 |
| | FC Dropout | 0.1, 0.25, 0.5 |
| | FC Norm | none, batch, layer |
| | FC Activation | ReLU, LeakyReLU, PReLU |
| | GCN Classifier Layers | 1, 2, 4 |
| | Enriching Edge Layers | 2, 4 |
| Discriminator | Embedded Dim. | 16, 32, 64, 128, 256 |
| | GCN Architecture | (a) 1 $\frac{1}{2}$ $\frac{1}{4}$ $\frac{1}{8}$ $\frac{1}{8}$ $\frac{1}{8}$ |
| | | (b) 1 1 $\frac{1}{2}$ $\frac{1}{2}$ $\frac{1}{4}$ $\frac{1}{4}$ $\frac{1}{8}$ $\frac{1}{8}$ $\frac{1}{8}$ |
| | | (c) 1 2 2 1 1 $\frac{1}{2}$ $\frac{1}{2}$ $\frac{1}{4}$ $\frac{1}{4}$ $\frac{1}{8}$ $\frac{1}{8}$ $\frac{1}{8}$ |
| | Update Every | 1, 5, 10, 50, 100, 200, 500, 1000, 2000 |

Table 6: *Quantitative Results*, including our defined measures for the scene graph enrichment task, are exhibited for selected hyperparameter settings.

| Measure/HP | SG Enrich | Deep G | Best NoD | 128 BS |
|---|---|---|---|---|
| Obj. Acc. | 19.04 | 17.31 | 15.09 | 16.85 |
| Avail. Pred. Acc. | 41.95 | 39.90 | 40.25 | 31.38 |
| Not Avail. Pred. Acc. | - | - | 3.02 | - |
| Avail. Edge Acc. | 75.00 | 75.64 | 76.08 | 66.89 |
| Not Avail. Edge Acc. | 99.58 | 99.70 | 99.54 | 99.54 |
| Scene Class. Acc. | 20.33 | 20.84 | 19.18 | 25.33 |
| Mini-Batch Size | 32 | 32 | 32 | 128 |
| GConv Dropout | 0.1 | 0.05 | 0.05 | 0 |
| GConv Normalization | none | none | BN | none |
| GConv Activation | LeReLU | LeReLU | PReLU | ReLU |
| Obj. Loss Weight | 1000 | 500 | 1000 | 200 |
| A. Pred Loss Weight | 100 | 50 | 100 | 50 |
| N.A. Pred Loss Weight | 0.1 | 0.01 | 5 | 0.01 |
| Edge Loss Weight | 1 | 0.01 | 100 | 10 |
| GAN Loss Weight | 0.1 | 1 | 0 | 1 |
| Scene Features Loss Weight | 200 | 0.1 | 0.001 | 0.01 |
| Scene Features | $hpooled_{l_1}$ | $logit_{l_2}$ | $logit_{CE}$ | $logit_{CE}$ |
| G Embed. Dim. | 256 | 128 | 128 | 128 |
| G GCN Architecture | (a) | (e) | (c) | (a) |
| G FC Dropout | 0.1 | 0.1 | 0.5 | 0.5 |
| G FC Normalization | BN | none | LN | BN |
| G FC Activation | LeReLU | ReLU | ReLU | ReLU |
| G Classifier Layers | 2 | 2 | 2 | 1 |
| G Enriching Edge Layers | 2 | 4 | 2 | 2 |
| D Embed. Dim. | 16 | 32 | - | 16 |
| D GCN Architecture | (b) | (c) | - | (b) |
| D Update Every | 200 | 1 | - | 1 |

### B.3 Training Details & Hyperparameter Tuning

To show the iterative power of the enrichment process, the model is forced to enrich an object not already present in the input SG at each step during inference. Adam optimizer (Kingma & Ba, 2014) is used with $10^{-4}$ as the learning rate. Also, We employed early termination for training our network based on the accuracies of the validation split. Using the proposed model on the VG dataset, continuing the training leads to improvement in training metrics and worsening validation metrics, indicating overfitting.

Table 5 describes the various settings used for hyperparameter tuning. Before training the model end-to-end, a hyperparameter optimization is performed, involving only our SG generator and discriminators. In other words, no image synthesizer is used at that phase. That is because training and inference time are increased while disconnecting the two mentioned components. Accordingly, the underlined values in Table 5 are indeed eliminated from the final hyperparameter tuning for the end-to-end model, as they performed poorly in the first stage. As already mentioned, loss weights have a crucial impact on the performance of our model. For instance, assigning a higher weight to *Not Avail. Pred.* could result in a model that outputs an enriching object without any enriching edges.

Hyperparameter tuning is performed as a sweep of different values described in Table 5. Best models are picked based on a better summation of validation accuracies in the model. Among them, the ones that performed qualitatively better on a limited selection of SGs are chosen. As so many possible combinations of hyperparameters result in a huge space to discover, we select a limited combination of the parameters based on Bayes hyperparameter optimization as we train around 650 different instances.

The mini-batch size varies from 32 to 128 scene graphs. A too-low value for mini-batch size leads to the GPU memory and processing power not being fully utilized. A too-high value for it results in longer training time as the GT values for each SG direct the weights updates in so many different directions. This may result in a poor update at each mini-batch iteration. Dropout layers are not used in the GCN version of Johnson et al. (2018), as the authors reported that using them reduces performance. Our experiments also show the same result for higher probability values of dropout. However, we found some lower values to be beneficial for overfitting prevention. Normalization layers are not used in (Johnson et al., 2018), but they potentially could aid the model to converge faster. As a result, we tried out Batch Normalization (Ioffe & Szegedy, 2015), Layer Normalization (Ba et al., 2016), or no normalization layer at all for our graph convolutions.

Too small values for the object loss weight or available predicates loss weight are eliminated based on our preliminary experiments, as these are two of the fundamentals of our overall objective. The option to set a weight of zero to *Not Avail. Pred.* loss is available to give the network the opportunity to update its weights based on only available predicates. In other words, when there is no edge between two objects in a scene graph, that does not mean the only possible reality is that these two do not have a relationship in a scene. Suppose the *Enriching Edge Detector* selects an edge where the predicate is unavailable in the GT. In that case, we may force the *Enriching Predicate GCN* to classify that edge as *none_pred*, indicating *no relationship* based on the GT or not penalize the output predicate as the GT is indeed unavailable. This is controlled using different values for predicate loss weights.

Additionally, a zero weight option for the GAN loss is considered, resulting in no discriminator and, thus, no adversarial loss for the model. When the GT and enriched SG pair is fed to the image generator, a pair of images are synthesized, which are the inputs of the *Visual Scene Characterizer*. For the pair of scene features, we considered them to be selected from one of the three sources, which are the final logits affiliated with the 365 scene categories, hidden scene features before the final layer of the network, and their average pooled version. For the scene logits, three cases are considered, which are $L_1$ or $L_2$ differences and cross-entropy loss. On the other hand, for the other two scene feature types, only $L_1$ or $L_2$ differences are assessed.

The embedded dimmension $D_{emb}$ for the generator is indeed the first GConv's $D_{in}$ selected from 16 to 256. $x_1\ x_2\ ... x_n$ architecture for GCN means there are $n-1$ GConv components stacked together where the $i^{th}$ GConv's input, output, and hidden dimensions are $x_i D_{emb}$, $x_{i+1} D_{emb}$, and $2(x_i + x_{i+1})D_{emb}$, respectively.

Architectures for Discriminators' GCNs are selected among the three available options. This architecture is, in fact, for the global discriminator, and for the local discriminator, the last two graph convolution layers are eliminated to have a shallower version.

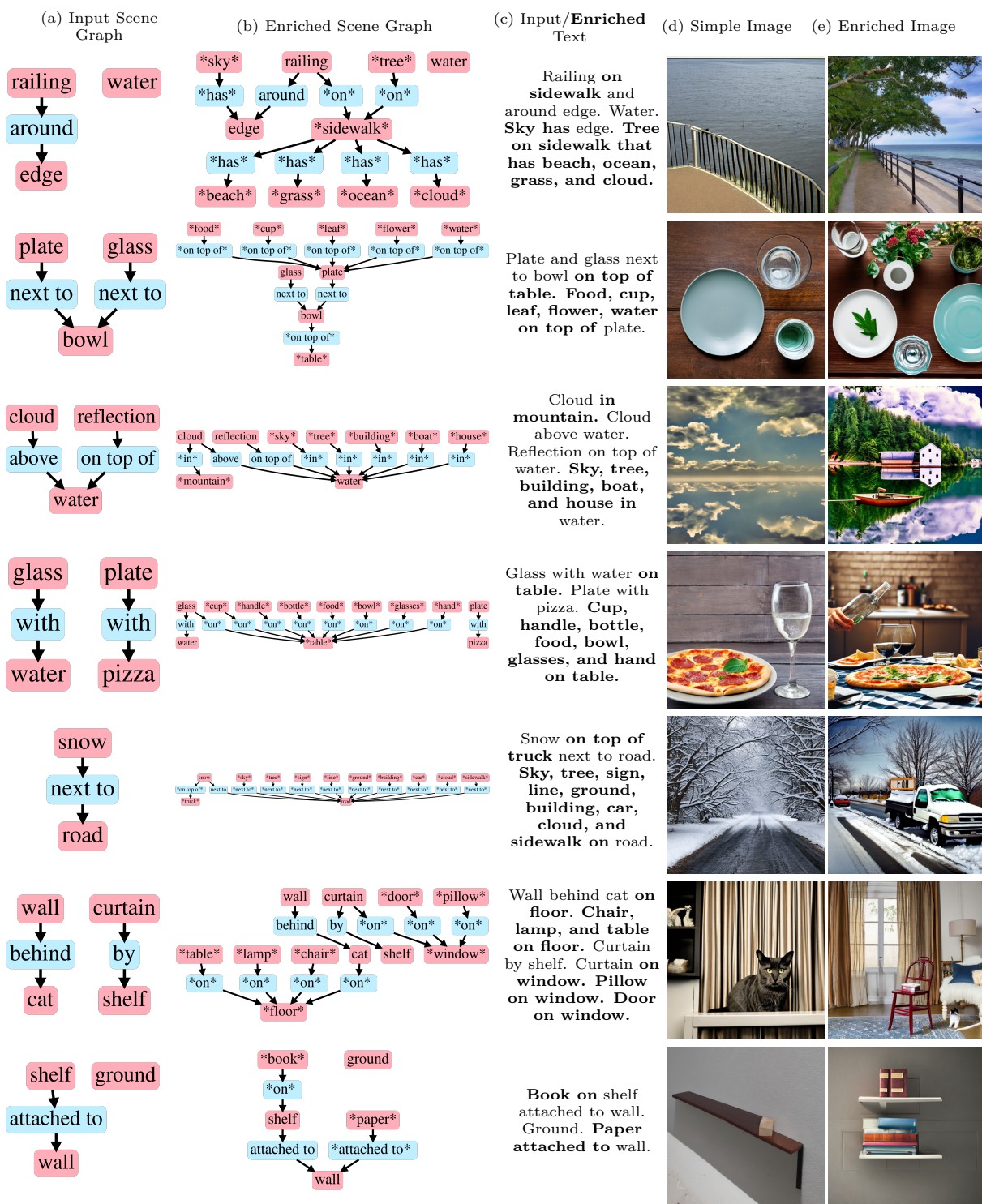

Figure 6: *Qualitative Results.* Samples of Visual Genome scene graphs and their enriched descriptions along with their corresponding synthesized images.

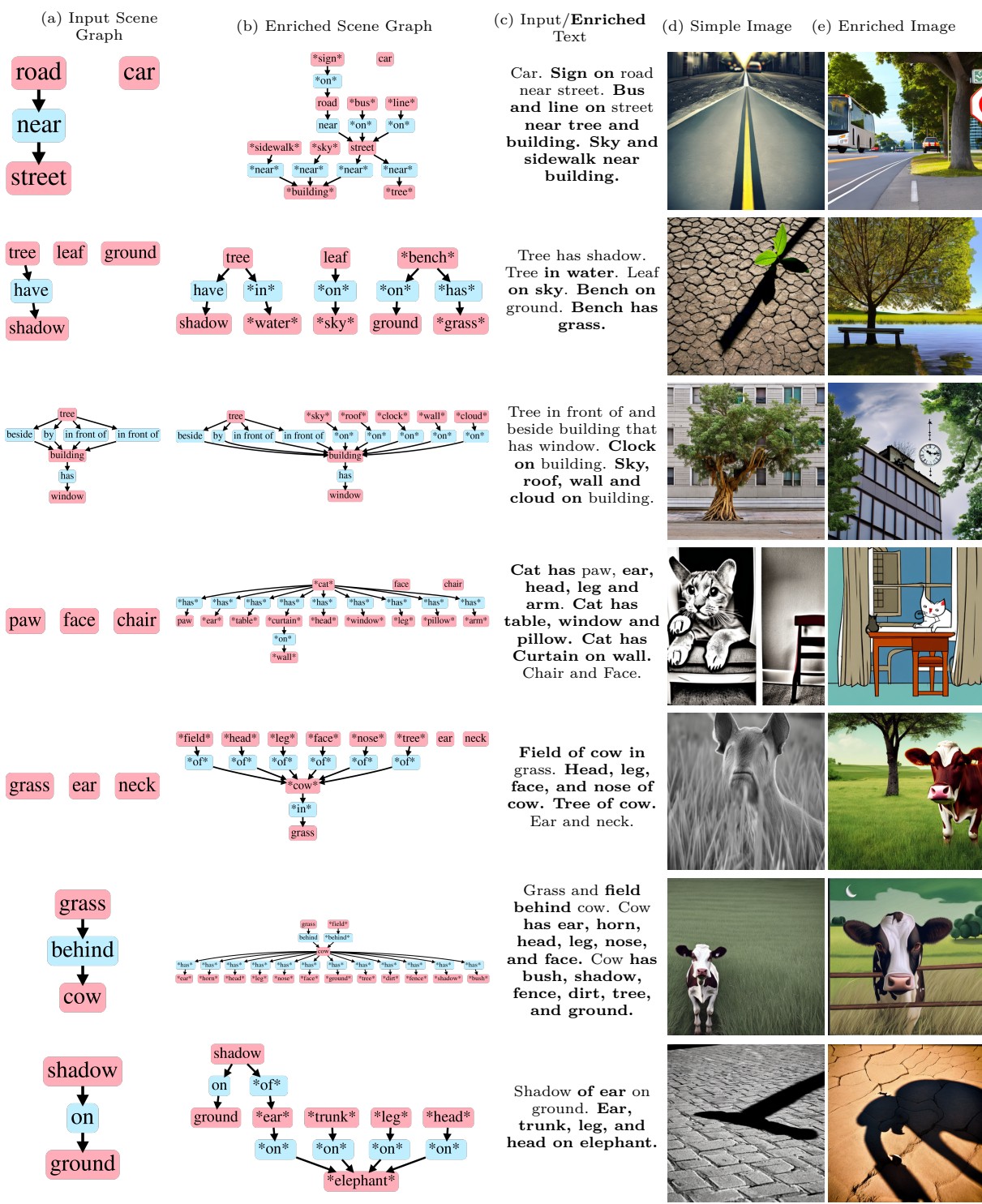

Figure 7: *Qualitative Results,* similar to Fig. 6

