# OpenReview forum: "Generated Contents Enrichment"
_TMLR — Rejected by TMLR_

### Review · Reviewer_dd9r · 2024-12-04

**Summary Of Contributions:**

The paper propose a novel approach to enrich input prompts to downstream image generation models by representing the input as a Scene Graph (SG). The input graph is enriched with additional object/subject nodes which are then connected to the existing scene graph via predicate edges in an iterative manner. The enrichment process is carried out via a graph convolution neural network trained to optimize different objective functions jointly. These entail an adversarial loss, through an appropriate discriminator/critic network, plus additional components which try to quantify the alignment between the image generated using the input scene graph and that generated using the enriched scene graph. Qualitative results show the generated images are richer than those obtained with the non-enriched scene graph and more coherent with respect to those generated by directly enriching the input prompt using a Large Language Model. Quantitative results align with the qualitative ones.

**Audience:**

Yes

**Broader Impact Concerns:**

I don't have particular concerns on the broader impacts of the work.

**Claims And Evidence:**

Yes

**Requested Changes:**

* I would add some additional baselines to compare to. Comparing with \[2\] seems reasonable to me, even though other approaches which I am not aware of might exist. Otherwise I would like to have a clarification on the arguments that prevent such comparison.
* The title could be more informative, "Generated Content Enrichment" seems to hint at the enrichment of some already generated content, while the paper proposes enriching the input prompt to produce richer content as a downstream result.
* I would clarify wether the quantitative results are the average of multiple runs (and if that is the case, how many and which is the standard deviation), otherwise, if they are the best results I would mention it.
* Most of the introduced symbols lack a clear specification of the respective dimensionalities, which makes it difficult to understand them clearly. I would add proper specification of this.
* It is unclear to me wether the Stable Diffusion used to generate the images is pre-trained on the specific dataset used in the experiments or not. Moreover, how is it adapted to use scene-graphs as conditioning inputs? Are they simply transformed to a textual description?
* Some of the claims made in the text seem too strong with respect to the proposed approach, I would rewrite them to align better with the paper's context. In particular:
	* Sec.1 Par.5: *"Our approach guarantees that the final enriched image accurately reflects the key scene elements specified in the input."* I would refrain from saying that a generative network has any guarantees of that if not formally proven.
	* Sec 3.6, after Eq. 14: *"This constraint guarantees that the predicted scene classes are identical"*. I would use a more appropriate wording, as I would not say there is any guarantee that the trained model satisfies such constraints.
	* Sec 5: *"We guarantee that the final enriched image accurately reflects the essential scene elements specified in the input description."* I would refrain from using strong terms as guarantee. The human subjects study itself disproves this claim by showing that 30% of the subjects think the enriched image does not reflect the description.
* The paper itself is not very formal in explaining the proposed architecture and the wording is often loose. Most of the explanation is verbose and remains blurry in the end. I would advise for a better organization of the content, trying to avoid redundancy while explaining the relevant aspects more concisely and directly. Some concepts from the Appendix can be merged in the main text, or the main text might remain on a higher level and precise architectural details can be moved and expanded in the Appendix. Overall I noticed the following problems:
	* Sec.1 Par.3: *"Secondly, the enriched scene graph, composited by the description and its enriched semantics, should be structurally real, following the distribution of scene graphs from real-world images."* The concept of scene graph has not yet been introduced.
	* Sec.1 Par.4: *"The third stage simultaneously feeds the enriched image and the input description into the Visual Scene Characteriser and Image-Text Aligner, thus the generated image is visually real and preserves the information of the description."* Given VSC and ITA are not yet introduced, it is unclear why the generated image should be visually realistic.
	* Sec.3: The division in subsections could be improved, there are many 3rd level subsections which could be replaced by paragraphs, moreover there are many empty 2nd level subsections, which serve just as wrappers for other subsections.
	* Sec.3.2: I find the explanation of the GCN hard to follow. In particular Eqs. 2 and 3 are unintuitive in my opinion. I would strive for an easier to read formulation of the graph convolution. Moreover, I would split Eqs. 2 and 3 in 4 separate lines. Right now it is unclear to me why one would refer separately to the two lines as they are currently arranged.
	* Sec.3.3.1: The way in which the Enriching Object network works is not completely clear. The authors state the implementation is that of the proposed GCN (Eq. 2 and 3), however, it is unclear, until later, how the enriching object is produced. I would try to unify the explanation to make it more understandable.
	* Sec.3.3.2: The Scene Graph Critic is explained very informally. *"GCN is employed to transform the input into another graph"*, *"input into additional neural network layers for further processing and discrimination"*. In general I think phrases could be less verbose and provide a more precise definition of what the network is doing.
	* Sec.4.5, Par. 4: *"Although the individual enriched subgraphs may seem realistic in W/o $D_{sg,glob}$, it is incapable of capturing the overall inconsistency between the enriched content and the entire scene."* This phrase is unclear, which evidence is backing it up? Is it just an intuition?
	* A.1: The explanation of the proposed GConv is note very clear. Many things regarding the architecture are explained loosely, defeating the purpose of having a section in the appendix to clarify such aspects.
	* A.3: The section is titled "Generator's GCN Architecture", but it goes on discussing only the Enriching Edge Detector, I would name it more appropriately. Moreover, the phrase *"The number of neural network layers in the GCN classifiers and Enriching Edge Detector model are selected via GCN Classifier Layers and Enriching Edge Layers in Table 5."* is not very clear. Do you mean that the number is the same or that it is a function of the number of layers in the other parts of the architecture?
	* B.3 Par.2: *"That is because training and inference time are increased while disconnecting the two mentioned components."*. This seems unintuitive to me, I would imagine the opposite to be true.
	* B.3 Par.4: *"A too-high value for it results in longer training time as the GT values for each SG direct the weights updates in so many different directions."*. What is the meaning of "many different directions"?
	* B.3 Par.6: *"For the pair of scene features, we considered them to be selected from one of the three sources, which are the final logits affiliated with the 365 scene categories, hidden scene features before the final layer of the network, and their average pooled version."*. This phrase could be improved as it is very difficult to understand.
	* B.3 Par 6 and 7: these two paragraphs seem out of context and seem more appropriate for the section regarding the model architecture.

**Strengths And Weaknesses:**

**Strenghts**
* The proposed approach is novel and tries to tackle a problem with under-specified prompts that might lead to uninteresting images being generated.
* The paper shows that the proposed method can be more effective in enriching inputs as it is more aligned with this specific task compared to a generic large language model.

**Weaknesses**
* It is not completely clear wether the problem the authors are trying to solve cannot be solved by a specialized large language model as there is no comparison with text-to-text models fine-tuned for the task at hand. While a study such as [1] might be considered concurrent work, it might be interesting to see how the strategies compare.
* While the proposed approach and related task might be considered formally novel, it seems there is a lack of comparison with other reasonable baselines which, given the ill-posedness of the problem and the lack of appropriate metrics, might help highligting the advantages of the proposed approach. In particular, the paper cites [2], but there is no actual comparison with it. From my understanding, while the approaches are different in spirit, they try to solve a similar enrichment problem. In fact, in [2], there are exmpirical results on using the enriched graph as prompt for image generation.
* The paper, while generally providing an intuitive explanation of the proposed task, which is easy to follow and understand, falls short when it comes to properly explaining the details of the proposed network, which most of the times end up being blurry or confusing/difficult to follow.

**Questions**
* It is unclear wether some form of randomness is fed as input to the generator or if it is completely determininstic given some input scene graph. Could you elaborate on this?
* Is there any reason why in Equations 13 and 15 two different measures of similarity are used?
* Why is the Image-Text Aligner considered as an additional component rather than a subtracted one in the ablation study?
* Did you explore the model's behavior when asked to enrich the graph more and more? From my understanding, it is still a manual task to define how many objects we need to enrich the original graph with. As text based models can enrich the description in one shot, is there a way to atuomatically stop enrichment in the proposed model?
* In Appendix A.1 is there a reason why the first FC layer has output shape 2*(Din + Dout)?

**References**
* \[1\] Zhan, Jingtao, et al. "Prompt Refinement with Image Pivot for Text-to-Image Generation." _arXiv preprint arXiv:2407.00247_(2024).
* \[2\] Agarwal, Rishi, et al. "GEMS: Scene expansion using generative models of graphs." _Proceedings of the IEEE/CVF Winter Conference on Applications of Computer Vision_. 2023.

---

> ### Author Response · Authors · 2025-03-25
> **Appreciation for Your Valuable Feedback**
>
> Thank you for your time and effort in providing valuable feedback on our work. We have carefully considered your comments and addressed each of the points raised below.

---

> ### Author Response · Authors · 2025-03-25
> **W1**
>
> **W1. It is not completely clear wether the problem the authors are trying to solve cannot be solved by a specialized large language model as there is no comparison with text-to-text models fine-tuned for the task at hand. While a study such as [1] might be considered concurrent work, it might be interesting to see how the strategies compare**
>
> We appreciate the suggestion and acknowledge the relevance of comparing with fine-tuned text-to-text models. However, our focus is on enriching scene graphs rather than textual descriptions. While we did not compare with an LLM specifically fine-tuned for this task, we conducted a thorough comparison with ChatGPT using three different prompting styles (Section 4.2, Figure 3, Table 2). As discussed in the paper, using LLMs for GCE presents several challenges: (1) training LLMs jointly with image generators is computationally expensive, (2) LLMs cannot evaluate the visual implications of enriched text, and (3) direct use of LLMs can lead to redundant content unsuitable for image generation. Our results (Table 2) highlight the limitations of LLM-based enrichment compared to our approach.

---

> ### Author Response · Authors · 2025-03-25
> **W2**
>
> **W2. While the proposed approach and related task might be considered formally novel, it seems there is a lack of comparison with other reasonable baselines which, given the ill-posedness of the problem and the lack of appropriate metrics, might help highligting the advantages of the proposed approach. In particular, the paper cites [2], but there is no actual comparison with it. From my understanding, while the approaches are different in spirit, they try to solve a similar enrichment problem. In fact, in [2], there are exmpirical results on using the enriched graph as prompt for image generation**
>
> Comparing with GEMS [2] is not meaningful, as our approaches target different tasks. GEMS expands a seed scene graph for image retrieval by generating novel scenes in the language domain, whereas our method enriches scene graphs explicitly for image synthesis. Unlike GEMS, our approach ensures that added objects and relationships enhance image quality and coherence. Although both models respect object co-occurrence patterns, our focus is on structural and semantic consistency in generated images.

---

> ### Author Response · Authors · 2025-03-25
> **Q1**
>
> **Q1. It is unclear wether some form of randomness is fed as input to the generator or if it is completely determininstic given some input scene graph. Could you elaborate on this**
>
> As shown in Figure 2, Stage 1 (scene graph enrichment) is deterministic, producing a probability distribution over potential enrichments rather than a single output. However, Stage 2 (image generation) introduces randomness, as image synthesis from a scene graph is inherently stochastic.

---

> ### Author Response · Authors · 2025-03-25
> **Q2**
>
> **Q2. Is there any reason why in Equations 13 and 15 two different measures of similarity are used**
>
> These losses capture different aspects of alignment. The scene classifier loss (Eq. 13) ensures the enriched image retains high-level scene characteristics consistent with the image generated from the original input. In contrast, the Image-SG Alignment Loss (Eq. 15) directly compares the enriched image with the input description, ensuring textual and visual coherence.

---

> ### Author Response · Authors · 2025-03-25
> **Q3**
>
> **Q3. Why is the Image-Text Aligner considered as an additional component rather than a subtracted one in the ablation study**
>
> The "SG Enrich" configuration was used during hyperparameter tuning due to computational constraints, whereas the final model ("With $M_{im\\_sg}$" in Table 3) includes the Image-Text Aligner. This setup ensures consistency across Tables 1 and 2 while emphasizing that the final results incorporate this module.

---

> ### Author Response · Authors · 2025-03-25
> **Q4**
>
> **Q4. Did you explore the model's behavior when asked to enrich the graph more and more? From my understanding, it is still a manual task to define how many objects we need to enrich the original graph with. As text based models can enrich the description in one shot, is there a way to atuomatically stop enrichment in the proposed model**
>
> Our method does not predefine the number of iterations; enrichment stops manually based on the image generator’s capacity. Even LLMs struggle with generating overly complex prompts for Stable Diffusion, which is why we constrained ChatGPT’s text generation in our comparison.

---

> ### Author Response · Authors · 2025-03-25
> **Q5**
>
> **Q5. In Appendix A.1 is there a reason why the first FC layer has output shape 2*(Din + Dout)**
>
> This generalization follows the architecture of SG2IM’s GCN model. We maintain this input-output pattern, as it has been empirically validated in prior work.

---

> ### Author Response · Authors · 2025-03-25
> **R1**
>
> **R1. I would add some additional baselines to compare to. Comparing with [2] seems reasonable to me, even though other approaches which I am not aware of might exist. Otherwise I would like to have a clarification on the arguments that prevent such comparison**
>
> Please refer to our response to W2.

---

> ### Author Response · Authors · 2025-03-25
> **R2**
>
> **R2. The title could be more informative, "Generated Content Enrichment" seems to hint at the enrichment of some already generated content, while the paper proposes enriching the input prompt to produce richer content as a downstream result**
>
> Thank you for the suggestion. We will refine the title in the final revision.

---

> ### Author Response · Authors · 2025-03-25
> **R3**
>
> **R3. I would clarify wether the quantitative results are the average of multiple runs (and if that is the case, how many and which is the standard deviation), otherwise, if they are the best results I would mention it**
>
> Results in Tables 1 and 3 are from single runs, as multiple executions with the same hyperparameters showed minimal variance. Table 2 follows standard practice: IS includes standard deviation, while FID is computed once over a large set (~40k–90k images) (Section 4.4.2).

---

> ### Author Response · Authors · 2025-03-25
> **R4**
>
> **R4. Most of the introduced symbols lack a clear specification of the respective dimensionalities, which makes it difficult to understand them clearly. I would add proper specification of this**
>
> We appreciate this observation and will clarify dimensional specifications in the final revision.

---

> ### Author Response · Authors · 2025-03-25
> **R5**
>
> **R5. It is unclear to me wether the Stable Diffusion used to generate the images is pre-trained on the specific dataset used in the experiments or not. Moreover, how is it adapted to use scene-graphs as conditioning inputs? Are they simply transformed to a textual description**
>
> Stable Diffusion was not pre-trained on VG. Scene graphs were converted to textual descriptions by transforming triplets (object1, predicate, object2) into textual statements such as "object1 predicate object2", as illustrated in Figures 6 and 7 of the Appendix.

---

> ### Author Response · Authors · 2025-03-25
> **R6**
>
> **R6. Some of the claims made in the text seem too strong with respect to the proposed approach, I would rewrite them to align better with the paper's context**
>
> We acknowledge this and will revise the wording to align more accurately with our findings.

---

> ### Author Response · Authors · 2025-03-25
> **R7**
>
> **R7. The paper itself is not very formal in explaining the proposed architecture and the wording is often loose. Most of the explanation is verbose and remains blurry in the end. I would advise for a better organization of the content, trying to avoid redundancy while explaining the relevant aspects more concisely and directly. Some concepts from the Appendix can be merged in the main text, or the main text might remain on a higher level and precise architectural details can be moved and expanded in the Appendix. Overall I noticed the following problems:**
>
> We are truly grateful that you took the time to read our paper in such detail and provide such delicate, insightful, and useful comments. Your feedback is invaluable, and we will refine the organization, reduce redundancy, and improve clarity in the final revision.

---

> ### Author Response · Authors · 2025-03-25
> **R7.1**
>
> **R7.1. Sec.4.5, Par. 4: "Although the individual enriched subgraphs may seem realistic in W/o, it is incapable of capturing the overall inconsistency between the enriched content and the entire scene." This phrase is unclear, which evidence is backing it up? Is it just an intuition**
>
> This statement is based on qualitative observations, not quantitative results. We will clarify this distinction.

---

> ### Author Response · Authors · 2025-03-25
> **R7.2**
>
> **R7.2. Moreover, the phrase "The number of neural network layers in the GCN classifiers and Enriching Edge Detector model are selected via GCN Classifier Layers and Enriching Edge Layers in Table 5." is not very clear. Do you mean that the number is the same or that it is a function of the number of layers in the other parts of the architecture**
>
> Yes, the number is the same.

---

> ### Author Response · Authors · 2025-03-25
> **R7.3**
>
> **R7.3. B.3 Par.4: "A too-high value for it results in longer training time as the GT values for each SG direct the weights updates in so many different directions.". What is the meaning of "many different directions"**
>
> This refers to conflicting gradient updates when using a large mini-batch. High variance in ground truth (GT) scene graphs leads to diverse gradient directions, slowing convergence and reducing generalization.

---

> > ### Comment · Reviewer_dd9r · 2025-03-27
> >
> > I thank the authors for their clarifications, however some concerns still remain.
> >
> > >**W1** [...] As discussed in the paper, using LLMs for GCE presents several challenges: (1) training LLMs jointly with image generators is computationally expensive, (2) LLMs cannot evaluate the visual implications of enriched text, and (3) direct use of LLMs can lead to redundant content unsuitable for image generation. Our results (Table 2) highlight the limitations of LLM-based enrichment compared to our approach.
> >
> > I am not entirely convinced by the arguments that prevent such comparison. (1) LLMs can be fine-tuned for specific tasks instead of being trained from scratch. (2) Visual implications are implicitly contained in the text describing an image, so I don't see why a text-to-text model should be unable to evaluate them, in particular if fine-tuned for such task. Similarly for (3), I don't see why a text-to-text model should be unable to overcome the redundancy limitation if fine-tuned for the considered task. However I acknowledge that doing such comparison can involve a meaningful amount of work in the absence of existing text-to-text models trained for this task.
> >
> > > **W2** Comparing with GEMS [2] is not meaningful, as our approaches target different tasks. GEMS expands a seed scene graph for image retrieval by generating novel scenes in the language domain, whereas our method enriches scene graphs explicitly for image synthesis. Unlike GEMS, our approach ensures that added objects and relationships enhance image quality and coherence. Although both models respect object co-occurrence patterns, our focus is on structural and semantic consistency in generated images.
> >
> > As mentioned in my comment, I aknowledge the difference in spirit between the two works, yet I fail to see why the proposed comparison would not be meaningful. On the contrary, I think it would be an important baseline to show that the proposed approach does indeed lead to better results if compared to other methods to enrich a scene-graph. Looking at Section 4.2 and Fig.2 of the GEMS paper they use their model to expand a scene-graph and generate an image from it. Moreover they employ the same Visual Genome dataset and generate images using the same SG2IM image generator. Given this, I would argue that the comparison is a meaningful and straightforward one.
> >
> > >**Q1** As shown in Figure 2, Stage 1 (scene graph enrichment) is deterministic, producing a probability distribution over potential enrichments rather than a single output. However, Stage 2 (image generation) introduces randomness, as image synthesis from a scene graph is inherently stochastic.
> >
> > This seems contradictory: "scene graph enrichment is deterministic, producing a probability distribution". I would imagine that if a probability distribution is given as output then you have to sample from it before passing the input to the image generator. In this sense it would be stochastic, otherwise, if given a graph as input your model produces always the same graph as output then it would be deterministic. Can you clarify this?
> >
> > >**Q2** These losses capture different aspects of alignment. The scene classifier loss (Eq. 13) ensures the enriched image retains high-level scene characteristics consistent with the image generated from the original input. In contrast, the Image-SG Alignment Loss (Eq. 15) directly compares the enriched image with the input description, ensuring textual and visual coherence.
> >
> > This does not answer my question. To be more clear, I wanted to know if, given both Eq. 13 and 15 show a measure of distance between two feature vectors, there was a particular reason to use the mean absolute error in the former and the mean cosine distance in the latter.
> >
> > >**R3** Results in Tables 1 and 3 are from single runs, as multiple executions with the same hyperparameters showed minimal variance. [...]
> >
> > I would encourage the authors to provide mean and variance values for the metrics across different runs with different randomized seeds. Given the lack of strong metrics to define the performance at task it is difficult to draw conclusions from just a single run.
> >
> > >**R7** [...] we will refine the organization, reduce redundancy, and improve clarity in the final revision.
> >
> > I think the magnitude of refinement is significant, hence the way in which it is done plays a relevant role in the evaluation of the paper.
> >
> > >**R7.3** This refers to conflicting gradient updates when using a large mini-batch. High variance in ground truth (GT) scene graphs leads to diverse gradient directions, slowing convergence and reducing generalization.
> >
> > In principle large batch sizes should result in better gradient estimates. I would advise to find some reference supporting this statement, as it feels counterintuitive.

---

> ### Author Response · Authors · 2025-03-28
>
> We sincerely appreciate the time and effort you have taken to review our paper and engage in this discussion. Your insightful feedback is invaluable in improving our work.
>
> **W1:**
> You are absolutely right that an LLM fine-tuned for this task, with carefully curated data and training techniques, could potentially address these limitations. However, out-of-the-box LLMs like ChatGPT, even with optimized prompting, struggle to outperform our graph-based approach. Our method remains significantly more efficient in both training and inference, making it a more practical choice for this task.
>
> **W2:**
> We acknowledge that the GEMS paper is one of the most relevant works to ours. However, the comparison is not as straightforward as it may seem. While both approaches use the Visual Genome dataset, the core differences lie in:
>
> 1. Task Objective: GEMS focuses on image retrieval, while we focus on scene graph enrichment and scene graph-to-enriched-image synthesis.
>
> 2. Use of External Knowledge: GEMS explicitly incorporates language-based external knowledge, whereas our method operates within scene graphs during training.
>
> 3. Seed Graph Differences: The Visual Genome dataset does not provide predefined seed graphs. Both our method and GEMS rely on pruned graphs, but these pruning strategies may differ significantly, making direct comparisons unreliable.
>
> 4. Evaluation Methodology: Both approaches evaluate scene-graph-to-image quality using FID and IS. However, our results demonstrate significantly better FID and IS scores than those reported in GEMS, highlighting the effectiveness of our enrichment approach. Additionally, we leverage Stable Diffusion, which is considerably more advanced than SG2IM, further improving image synthesis quality.
>
> Since the GEMS implementation and its seed graphs are not publicly available, aligning the experimental settings for a fair comparison is infeasible. Without access to their exact methodology, we cannot reliably re-evaluate either approach under identical conditions.
>
> **Q1:**
> Our scene graph enrichment module is deterministic in the sense that, given the same weights and parameters, it will always produce the same enriched graph. However, within each iteration, it generates a probability distribution over possible enriching objects or predicates. This is akin to standard classification tasks, where a probability distribution is computed over possible classes, and the highest-probability outcome is selected.
>
> **Q2:**
> Apologies for the confusion. The difference in distance metrics arises from normalization. In Eq. 15, the feature vectors are always normalized, making cosine distance the appropriate choice. In contrast, Eq. 13 operates on potentially unnormalized features, where mean absolute error is more suitable.
>
> **R3:**
> This is a valuable suggestion, and we look into incorporating mean and variance values for Tables 1 and 3, where we compare different versions of our model. However, for FID to compare with other models (Table 2), multiple runs yield the same value unless the input images differ. In literature, FID is typically reported once because its variance stems from dataset size rather than randomness in model execution. Our goal is to compute FID over as many images as computationally feasible to ensure robustness.
>
> **R7.3:**
> We acknowledge that batch size effects can vary across models and tasks. In our case, we observed performance degradation with large mini-batches, likely due to conflicting gradient updates. High variance in ground truth scene graphs introduces diverse gradient directions, slowing convergence and reducing generalization. Similar findings are reported in prior work:
> - Keskar et al. [1] show that large batches converge to sharp minima, which harm generalization.
> - McCandlish et al. [2] identify an optimal batch size that minimizes gradient variance without instability. Increasing beyond this batch-size leads to diminishing returns.
> - Jastrzębski et al. [3] demonstrate that excessive variance reduction from large batches can worsen generalization.
>
> [1] Keskar, Nitish Shirish, et al. "On large-batch training for deep learning: Generalization gap and sharp minima." arXiv preprint arXiv:1609.04836 (2016).
> [2] McCandlish, Sam, et al. "An empirical model of large-batch training." arXiv preprint arXiv:1812.06162 (2018).
> [3] Jastrzębski, Stanisław, et al. "Three factors influencing minima in SGD." arXiv preprint arXiv:1711.04623 (2017).

---

### Review · Reviewer_Vgm4 · 2025-02-17

**Summary Of Contributions:**

The paper deals with graph conditioned image generative models. In particular, the paper introduces an approach to enhance the scene graph with an objective to improve the quality of image generation. The introduced approach leverages GCNs and GANs. The method is compared to ChatGPT enhanced text and simple graph conditioning method and is shown to outperform the baselines. The qualitative and quantitative results are supported with a small-scale user study.

**Audience:**

Yes

**Broader Impact Concerns:**

No need for broader impact statement.

**Claims And Evidence:**

No

**Requested Changes:**

Please address the weaknesses part.

Clarification questions:
How the scene graphs look like for the prompts displayed in Figure 3? Why are some parts of the prompt boldfaced? Is the input to the stable diffusion converted to text or the model is working on top of the graphs?

**Strengths And Weaknesses:**

Strengths: Overall, the paper deals with a problem that is of interest to the TMLR community. The proposed solution looks like a reasonable way to enhance image generation. Human evaluation is a nice addition to the paper.


Weaknesses:
- The overall presentation of the paper is not easy to get through. The paper could be trimmed, the captious should be more descriptive (self-explanatory, e.g. what is the metric used in ablations, Table 3).

- There are more recent scene-graph to image pipelines (e.g. https://arxiv.org/pdf/2211.11138) than Johnson et al., 2018. What is the reason that the work extends an older pipeline instead of starting from a more recent one that is already based on diffusion?

- The paper lacks comparisons to related work on VG dataset, without reporting prior art results it is hard to know if the numbers reported in table 2 are strong and makes it hard to assess the significance of the reported results. For example, how the FID numbers compare to the ones reported here: https://arxiv.org/pdf/2211.11138

- Second paragraph of the intro, “, barely investigated in conventional research,”. I would suggest rephrasing this statement as the use of LLMs to enhance input prompts is quite common among practitioners, thus, I would not say that GCE is “barely investigated”.

---

> ### Author Response · Authors · 2025-03-25
> **Appreciation for Your Valuable Feedback**
>
> Thank you for your constructive feedback. We have carefully addressed each of your concerns below.

---

> ### Author Response · Authors · 2025-03-25
> **W1**
>
> **W1. The paper could be trimmed, the captious should be more descriptive (self-explanatory, e.g. what is the metric used in ablations, Table 3)**
>
> The metrics in Table 3 are introduced in Section 4.4.1. We appreciate this suggestion and will add a concise description to the table caption to make it more self-explanatory.

---

> ### Author Response · Authors · 2025-03-25
> **W2**
>
> **W2. There are more recent scene-graph to image pipelines (e.g. https://arxiv.org/pdf/2211.11138) than Johnson et al., 2018. What is the reason that the work extends an older pipeline instead of starting from a more recent one that is already based on diffusion**
>
> We chose to build on the Graph Convolutional Network (GCN) approach from SG2IM (Johnson et al., 2018) rather than the diffusion-based SGDiff (https://arxiv.org/pdf/2211.11138) for efficiency reasons. Our method is already iterative, and incorporating an additional diffusion process within each iteration would significantly increase computational cost and reduce scalability. GCNs offer a more efficient alternative while still capturing the necessary semantic relationships.
>
> Furthermore, for image generation, we already employ Stable Diffusion for evaluation, which outperforms SGDiff in terms of image quality. As stated in the paper, our primary goal is to demonstrate the feasibility of enriching generated content rather than surpassing state-of-the-art text/graph-to-image models. Our method is modular and can be combined with any image generator, including SGDiff, to achieve enhanced performance.

---

> ### Author Response · Authors · 2025-03-25
> **W3**
>
> **W3. The paper lacks comparisons to related work on VG dataset, without reporting prior art results it is hard to know if the numbers reported in table 2 are strong and makes it hard to assess the significance of the reported results. For example, how the FID numbers compare to the ones reported here: https://arxiv.org/pdf/2211.11138**
>
> Our Scene Graph Enricher (Stage 1 in Fig. 2) is designed as an add-on that can enhance any image generator (Stage 2 in Fig. 2). In Table 2, the "Simple" column represents results from Stable Diffusion, a state-of-the-art image generation model that performs better than SGDiff (https://arxiv.org/pdf/2211.11138). The "Enriched Ours" column demonstrates that applying our enrichment method on top of Stable Diffusion improves results, showing its effectiveness independently of the underlying image generator.
>
> To further highlight our model’s impact, we also compare it to an alternative enrichment approach—using ChatGPT for text-based enrichment. The results in Table 2 show that our method outperforms ChatGPT-based enrichments, reinforcing the advantage of structured scene graph-based enrichment over purely text-based modifications.
>
> While direct FID comparisons to SGDiff are not provided, our method could be integrated with SGDiff in the same way it is applied to Stable Diffusion, enabling a direct comparison in future work.

---

> ### Author Response · Authors · 2025-03-25
> **W4**
>
> **W4. Second paragraph of the intro, “, barely investigated in conventional research,”. I would suggest rephrasing this statement as the use of LLMs to enhance input prompts is quite common among practitioners, thus, I would not say that GCE is “barely investigated”**
>
> We appreciate this suggestion and agree that the phrase "barely investigated in conventional research" could be misleading, given the growing use of LLMs for prompt enhancement. We will revise this statement accordingly.

---

> ### Author Response · Authors · 2025-03-25
> **Clarification questions**
>
> **Clarification questions: How the scene graphs look like for the prompts displayed in Figure 3? Why are some parts of the prompt boldfaced? Is the input to the stable diffusion converted to text or the model is working on top of the graphs**
>
> The full versions of the scene graphs are similar to those shown in Appendix Figures 6 and 7. In Figure 3, the (a) input description is generated by converting each triplet (object1, predicate, object2) from the scene graph into a simple text format: "object1 predicate object2."
>
> The boldfaced text in column (c) highlights the parts enriched by our model relative to the input description. Since Stable Diffusion only accepts text prompts, the enriched scene graphs are converted into textual descriptions using this same mechanism before generating the final images in Figure 2.

---

### Review · Reviewer_6EsQ · 2025-03-12

**Summary Of Contributions:**

- A new task, "Generated Contents Enrichment (GCE)," is introduced, focusing on generating content that is visually realistic, semantically abundant, and particularly structurally coherent.
- Addressing the challenge of GCE, this paper proposes a pipeline that expands simple descriptions into enriched scene graphs to generate enriched images, using scene graphs as a means to bridge the semantic richness gap between visual content and textual descriptions.
- Additionally, this paper introduces a Generative Adversarial SG Enrichment technique, designed to explicitly expand text semantics and their relationships, continuously enriching the scene graph content.

**Audience:**

Yes

**Claims And Evidence:**

Yes

**Requested Changes:**

- To enhance the evaluation metrics in Tables 1 and 3, it is crucial to select metrics that accurately reflect the validity and diversity of the scene graph.
- It is critical to include tests on external datasets to demonstrate that the method does not simply overfit to the training data distribution.
- It is crucial to revise the evaluation metrics in ablation studies to clearly identify the specific contributions of each module.
- It is better to clearly explain the scalability issues and the unique significance of the method under the context of currently relying on a vast amount of text-image pairs for training.

**Strengths And Weaknesses:**

Strengths:

- This paper introduces a new task that aims to generate visually rich and semantically dense content from language descriptions, with a particular emphasis on controllable structures.
- The method presented in this paper is straightforward, enhancing semantic richness by continually expanding the nodes and edges within the scene graph.

Weaknesses:

- Potential challenges and ambiguities in training: The training process for the Scene Graph Enricher involves sampling a ground truth scene graph, randomly deleting a node and its edges, and predicting the deleted node from the remaining nodes. This approach introduces considerable uncertainty due to the reliance on incomplete information and the possibility of multiple plausible predictions, especially in scenarios with limited data availability.
- The method in this paper remains limited by the quality of the image generator. Although it generates richer scene graphs, it does not extend the generative capabilities of the image generator, thus failing to produce visually richer semantic content.
- The evaluation metrics in Tables 1 and 3 seem insufficient to effectively reflect the validity and semantic richness of the enriched scene graphs generated by the Scene Graph Enhancer. Particularly, the near 100% performance across the board in Not Avail. Edge Acc. suggests that this metric may have limited significance for further evaluation. It would be more appropriate to adopt metrics that better represent the quality of scene graph enhancement to more accurately measure the model's effectiveness.
- The evaluation lacks generalization across varied scenarios. This study only tests on the VG dataset and does not include external datasets, naturally leading to improvements over an untrained baseline within the same domain. It is unclear whether these gains compromise generalization to general scenes.
- From the results of the ablation study, the specific contributions of each module are difficult to discern. In Table 3, the performance differences between all modules are minimal, and given the limitations of the current evaluation metrics, it is challenging to accurately assess the actual impact of each module.
- Concerns about scalability arise with the proposed method due to its reliance on scene graph annotations, which are difficult to obtain in large, high-quality quantities. The rapid progress in the generative domain has largely been fueled by the availability of extensive datasets. However, this method's dependency might limit its scalability, particularly with the complex and diverse images found online, where data acquisition costs and annotation quality become critical constraints.

---

> ### Author Response · Authors · 2025-03-25
> **Appreciation for Your Valuable Feedback**
>
> Thank you for your thoughtful and constructive feedback. We have carefully addressed each of your concerns below.

---

> ### Author Response · Authors · 2025-03-25
> **W1**
>
> **W1. Potential challenges and ambiguities in training: The training process for the Scene Graph Enricher involves sampling a ground truth scene graph, randomly deleting a node and its edges, and predicting the deleted node from the remaining nodes. This approach introduces considerable uncertainty due to the reliance on incomplete information and the possibility of multiple plausible predictions, especially in scenarios with limited data availability.**
>
> We acknowledge the inherent uncertainty in predicting a missing object from an incomplete scene graph. However, this is a common challenge in many machine learning models, including large language models (LLMs), which predict the next word from a probability distribution rather than a single correct answer. As discussed in Section 4.4.1, our approach similarly predicts a probability distribution over possible object categories rather than selecting a single deterministic output.
>
> To mitigate the limitations of individual metrics, we supplement our evaluation with image-level measures such as Inception Score (IS) and Fréchet Inception Distance (FID) (Table 2), as well as human evaluations (Table 4). Additionally, since the Visual Genome (VG) dataset inherently contains multiple plausible object continuations for similar contexts, our model naturally learns to handle variations in data distribution.

---

> ### Author Response · Authors · 2025-03-25
> **W2**
>
> **W2. The method in this paper remains limited by the quality of the image generator. Although it generates richer scene graphs, it does not extend the generative capabilities of the image generator, thus failing to produce visually richer semantic content.**
>
> The component you refer to as the "image generator" corresponds to Stage 2 in Figure 2. Our Scene Graph Enricher (Stage 1) is designed to be modular and can be integrated with any scene graph-to-image model to enrich semantics, as demonstrated qualitatively in Figure 3 (Simple vs. Enriched) and quantitatively in Table 2.
>
> While we do not modify the image generator itself, our loss functions—especially those in Section 3.5, which operate directly on the output images—ensure that the enriched scene graphs yield visually richer, structurally coherent, and semantically abundant images. This is reflected in the substantial improvements in our generated results.

---

> ### Author Response · Authors · 2025-03-25
> **W3**
>
> **W3. The evaluation metrics in Tables 1 and 3 seem insufficient to effectively reflect the validity and semantic richness of the enriched scene graphs generated by the Scene Graph Enhancer. Particularly, the near 100% performance across the board in Not Avail. Edge Acc. suggests that this metric may have limited significance for further evaluation. It would be more appropriate to adopt metrics that better represent the quality of scene graph enhancement to more accurately measure the model's effectiveness**
>
> We recognize that the metrics in Tables 1 and 3 alone may not fully capture the semantic richness of enriched scene graphs. Therefore, we complement them with image-level evaluations in Table 2 (FID, IS) and user studies in Section 4.6 to provide a more comprehensive assessment.
>
> Regarding the high performance of "Not Avail. Edge Acc.," this result is expected due to the nature of the metric. As discussed in Section 4.4.1, this metric evaluates the model’s ability to recognize the absence of relationships in the ground truth (GT). The high accuracy suggests that identifying "no relationship" is an easier task compared to predicting the presence of new connections as scene graphs are typically sparse graphs.
>
> We appreciate any suggestions for alternative scene graph enrichment metrics that go beyond object/predicate accuracy and better quantify the effectiveness of our method.

---

> ### Author Response · Authors · 2025-03-25
> **W4**
>
> **W4. The evaluation lacks generalization across varied scenarios. This study only tests on the VG dataset and does not include external datasets, naturally leading to improvements over an untrained baseline within the same domain. It is unclear whether these gains compromise generalization to general scenes**
>
> We extensively discuss dataset limitations in Section 4.1. The VG dataset is the most suitable available dataset for our task, as it contains detailed scene graphs with 110k real-world images, covering 178 object and 45 predicate categories.
>
> Alternative datasets, such as COCO-Stuff, lack explicit scene graph annotations and only provide coarse relationships derived from 2D coordinates (e.g., "left of" "above"), which limits their applicability to our task. Given the absence of better alternatives, we argue that VG provides sufficient variety to assess generalization across diverse scenes.

---

> ### Author Response · Authors · 2025-03-25
> **W5**
>
> **W5. From the results of the ablation study, the specific contributions of each module are difficult to discern. In Table 3, the performance differences between all modules are minimal, and given the limitations of the current evaluation metrics, it is challenging to accurately assess the actual impact of each module**
>
> The most informative metrics for evaluating individual module contributions in Table 3 are Obj. Acc., Avail. Pred. Acc., and Scene Class. Acc. Despite the seemingly small numerical differences, each module incrementally improves performance. For example, the removal of certain components results in a few percentage points of accuracy loss, which is meaningful given the complexity of the task.

---

> ### Author Response · Authors · 2025-03-25
> **W6**
>
> **W6. Concerns about scalability arise with the proposed method due to its reliance on scene graph annotations, which are difficult to obtain in large, high-quality quantities. The rapid progress in the generative domain has largely been fueled by the availability of extensive datasets. However, this method's dependency might limit its scalability, particularly with the complex and diverse images found online, where data acquisition costs and annotation quality become critical constraints**
>
> We acknowledge that large-scale, high-quality scene graph annotations are less available than textual data for LLMs. However, despite using significantly less data and compute than LLMs, our model still outperforms them in the Generated Content Enrichment (GCE) task, as demonstrated in Table 2.
>
> While an LLM could theoretically perform scene graph enrichment using textual descriptions alone, it faces several limitations:
> 1. Training inefficiency – LLMs require vast amounts of training data and compute to simultaneously model text and image generation.
> 2. Lack of visual grounding – While LLMs can enrich text, they do not ensure that the augmented descriptions align with the visual content of a generated image.
> 3. Redundant content generation – LLMs tend to produce overly complex descriptions that may not translate effectively into visual synthesis, a challenge even for state-of-the-art image generators.
>
> To empirically validate this, we compare our approach to ChatGPT-generated enrichments in Figure 3 and Table 2, showing that our method leads to more coherent and semantically meaningful enriched images.

---

### Decision · Action_Editor_Gqia · 2025-05-11

**Recommendation:** Reject

**Comment:**

While the reviewers generally considered that the paper addresses an interesting problem with a reasonable strategy, they raised several major concerns:
1. Technical clarity: the paper lacks sufficient clarity in presentation (dd9r, Vgm4) and the reviewers raised concerns on the technical design (6EsQ).
2. Insufficient experimental evaluation: the reviewers pointed out the experiments lack comparisons with fine-tuned LLMs and scene graph-based methods (dd9r, Vgm4), the metrics and dataset are insufficient (6EsQ), and its generalization and scalability are unclear (6EsQ).
3. Motivation: the reviewers pointed out overclaims of novelty or insufficient motivation on the problem setting (dd9r, Vgm4).

The author partially responded to some of these concerns, but did not fully addressed the reviewers' questions nor revised the manuscript. After the author response phase, all three reviewers remained unconvinced due to the aforementioned concerns and hence leaned toward or recommend rejection. Based on the review and discussion, the AE finds the reviewers' arguments compelling and no sufficient reason to overturn the reviewers' ratings. Overall, the efficacy of the proposed strategy remains unclear and the paper requires a significant revision to improve its presentation and evaluation before it is ready for publication.

**Audience:**

The topic of controllable image generation is of interest to the audience in machine learning and computer vision.

**Claims And Evidence:**

The paper introduces a novel task of conditional image generation, Generated Contents Enrichment (GCE), and proposes a scene graph-based approach that enriches input prompts in order to bridge the semantic richness gap between visual content and textual descriptions. In particular, it introduces an adversarial learning method to explicitly expand text semantics and their relationships iteratively based on a graph convolution network. The paper presents experimental evaluation on VisualGenome dataset with comparisons to multiple baselines.

The main claim lies at the novelty of the proposed technical approach and its advantages in generating visually realistic, structurally coherent, and semantically abundant images. While the paper shows promising results over the baselines, it falls short in providing convincing evidence due to 1) Lack of comparison with recent related methods; 2) Insufficient evaluation metrics and generalization evaluation; 3) Lack of clarity in presentation and technical details.